# Nr4a1 and Nr4a3 redundantly control clonal deletion and contribute to an anergy-like transcriptome in auto-reactive thymocytes to impose tolerance in mice

Hailyn V. Nielsen [1], Letitia Yang[2], James L. Mueller[1], Alexander J. Ritter [1], Ryosuke Hiwa [3], Irina Proekt [4], Elze Rackaityte [5], Dominik Aylard[6], Mansi Gupta [7], Christopher D. Scharer [7], Mark S. Anderson [4], Byron B. Au-Yeung [8] ✉ & Julie Zikherman [1] ✉

The Nr4a nuclear hormone receptors are transcriptionally upregulated in response to antigen recognition by the T cell receptor (TCR) in the thymus and are implicated in clonal deletion, but the mechanisms by which they operate are not clear. Moreover, their role in central tolerance is obscured by redundancy among the Nr4a family members and by their reported functions in Treg generation and maintenance. Here we take advantage of competitive bone marrow chimeras and the OT-II/RIPmOVA model to show that *Nr4a1* and *Nr4a3* are essential for the upregulation of *Bcl2l11*/BIM and thymic clonal deletion by self-antigen. Importantly, thymocytes lacking *Nr4a1/3* acquire an anergy-like signature after escaping clonal deletion and Treg lineage diversion. We further show that the Nr4a family helps mediate a broad transcriptional program in self-reactive thymocytes that resembles anergy and may operate at the margins of canonical thymic tolerance mechanisms to restrain self-reactive T cells after thymic egress.

Deletion of highly self-reactive thymocytes and diversion of others into the regulatory T cell lineage during their development are vital to prevent autoimmune disease. Indeed, mice and humans with defects in these central T cell tolerance mechanisms exhibit a broad range of autoimmune pathology[1–3]. Positive selection by specialized cortical thymic epithelial cells (cTECs) ensures MHC restriction among double positive (DP) thymocytes[4]. Although excessive recognition of self-pMHC in the thymic cortex can trigger deletion of DP thymocytes at this early stage, many important self-antigens are tissue-restricted[4–7]. Only after their migration to the thymic medulla following positive selection are thymocytes tolerized to such tissue-restricted antigens[4,5,7]. Promiscuous gene expression by AIRE-positive medullary thymic epithelial cells (mTECs) along with recently discovered thymic mimetic cells serves to eliminate self-reactivity from the conventional mature T cell repertoire[2,6,8,9]. In contrast to the thymic cortex, medullary self-antigen encounter can drive both Treg diversion and deletion[5]. However, the

[1]Division of Rheumatology, Rosalind Russell and Ephraim P. Engleman Arthritis Research Center, Department of Medicine, University of California, San Francisco, CA 94143, USA. [2]Biomedical Sciences Graduate Program, University of California, San Francisco, CA 94143, USA. [3]Department of Rheumatology and Clinical Immunology, Graduate School of Medicine, Kyoto University, 54 Kawahara-cho, Shogoin, Sakyo-ku, Kyoto 606-8507, Japan. [4]Diabetes Center, Department of Medicine, University of California, San Francisco, CA 94143, USA. [5]Department of Biochemistry and Biophysics, University of California, San Francisco, CA 94143, USA. [6]Department of Molecular & Cell Biology, University of California, Berkeley, CA 94720, USA. [7]Department of Microbiology and Immunology, Emory University, Atlanta, GA 30322, USA. [8]Division of Immunology, Lowance Center for Human Immunology, Department of Medicine, Emory University, Atlanta, GA 30322, USA. ✉e-mail: byron.au-yeung@emory.edu; julie.zikherman@ucsf.edu

molecular pathways that direct these fate decisions are incompletely understood. Although the pro-apoptotic BH3-only family member BIM is implicated in the deletion of self-reactive thymocytes[10], the mechanisms linking it to self-antigen recognition are not clear. In contrast to the discrete fates of death and Treg diversion, TCR affinity for a given self-antigen exists on a spectrum, highlighting the need for a cell-intrinsic restraint on clones with a TCR affinity just below the signaling threshold for death or diversion.

Among the few factors directly implicated in thymic negative selection are the Nr4a family of orphan nuclear hormone receptors. Nur77, Nurr1, and Nor1 (encoded by *Nr4a1, Nr4a2, and Nr4a3* respectively) are transcriptional regulators with conserved DNA-binding domains and shared binding site motif, but lack well-defined ligands. Rather, *Nr4a1-3* are primary response genes that are rapidly induced by antigen receptor signaling and are thought to be regulated at the level of their dynamic expression. They were first identified as mediators of antigen-induced cell death in T cell hybridomas, and studies of full-length and truncated dominant negative transgenes have suggested roles in thymic negative selection[11–15]. However, the relatively subtle phenotypes of *Nr4a1* and *Nr4a3* single knock-out mice raised the possibility that redundancy among Nr4a family members masks their role in thymic selection[16–20]. Conversely, mice lacking both *Nr4a1* and *Nr4a3* in the germline or T cell lineage exhibit loss of Treg with resultant severe immune dysregulation and death by 4 weeks of age[21–24]. Therefore, dissecting the roles of the Nr4a family during thymic negative selection and the mechanisms by which they operate has been technically challenging.

To overcome this challenge, we recently used a competitive chimera strategy to unmask redundant roles of *Nr4a1* and *Nr4a3* in tolerance, with a normal Treg compartment of WT genetic origin[21]. We found that polyclonal double knockout (DKO) thymocytes lacking both *Nr4a1* and *Nr4a3* exhibit a profound competitive advantage between DP and SP stages of thymic development, suggesting escape from negative selection. However, the anatomical site and developmental stage where Nr4a-dependent deletion occurs, as well as the molecular mechanisms involved, remain unclear. Indeed, both genomic and non-genomic Nr4a-dependent pathways to apoptosis have been proposed, but transcriptional targets that mediate deletion are unknown[25–27].

Here, we show that self-reactive DKO SP thymocytes fail to upregulate *Bcl2l11* transcript and BIM protein in response to self-antigen recognition and are completely rescued from AIRE-dependent negative selection. We further show that the Nr4a family helps to impose a broad anergy-like transcriptional program in semi-mature thymocytes that depends on reactivity to self-Ag in a TCR Tg model and scales with *Nr4a1* expression in the normal thymic repertoire, reflecting strong TCR signaling. Importantly, it persists among peripheral DKO T cells that escape thymic deletion but is also evident among WT recent thymic emigrants in the polyclonal, self-tolerant repertoire. The Nr4a family, therefore, mediates thymic deletion, Treg diversion, and contributes to a cell-intrinsic tolerance program initiated by self-Ag recognition in the thymus that may serve as a fail-safe mechanism to prevent autoimmunity.

## Results
### TCR signaling upregulates Nr4a family in thymocytes destined for deletion and Treg diversion

*Nr4a* transcripts are rapidly but transiently induced following Ag stimulation[28,29]. *Nr4a1* transcript is much more abundant than *Nr4a2* and *Nr4a3* in the T cell lineage (Supp Fig. 1a; Immgen.org). Mass spectrometry confirms that all three family members are rapidly upregulated in mature CD8+ cells by TCR stimulation, yet only Nr4a3 protein persists without decline over 24 hours, implying a longer protein half-life and suggesting levels may accumulate despite low abundance transcript

(Supp Fig. 1b; immpres.co.uk). Conversely, although Nr4a1/Nur77 protein is most highly induced, it is rapidly degraded[30–32].

Reporters of *Nr4a* transcript expression, including Nur77/*Nr4a1*-GFP BAC Tg, revealed sensitivity to affinity, dose, and duration of TCR stimulation and upregulation at signal-dependent checkpoints during thymic development[28,29,33–37]. Nur77-GFP is upregulated in post-selection DP thymocytes concurrently with CD69 induction, increases at the semimature SP stage[38], and is highest in agonist-selected Treg cells – consistent with their high degree of self-reactivity (Fig. 1a, b, Supp Fig. 1c, d).

*Nr4a1* and *Nr4a3* transcripts are upregulated in thymocytes undergoing negative selection[39,40]. Most recently, a CITEseq atlas of thymic development captured two clusters of thymocytes destined for deletion (Supp Fig. 1e)[41,42]. One of these clusters corresponds to CD69hi DP thymocytes with low *Ccr7* transcript levels and marks an initial wave of deletion triggered by excessive self-pMHC recognition in the cortex prior to CCR7-dependent migration to the medulla[43–45]. The second cluster is closely apposed to Foxp3-expressing Treg and captures CD69hi semimature SP thymocytes with high *Ccr7* transcript levels, likely those undergoing deletion in response to promiscuous expression of tissue-restricted antigens (TRA) in the medulla (Supp Fig. 1e)[6,46,47]. Notably, *Nr4a1* and *Nr4a3* transcripts are highly correlated with both deleting clusters, while *Nr4a2* is detectable in only a sparse number of cells (Supp Fig. 1e).

### Nr4a-deficient thymocytes exhibit a competitive advantage during development

We recently developed a competitive radiation chimera strategy in which donor bone marrow (BM) lacking both *Nr4a1* and *Nr4a3* (DKO) is mixed with an excess of WT BM to reconstitute a normal Treg compartment of WT origin (Fig. 1c)[21]. Chimeras reconstituted with a congenically marked WT:WT donor BM mix with matched input ratio serve as controls.

DKO cells exhibit a large cell-intrinsic advantage relative to WT among SP4 and SP8 thymocytes in the context of normocellular chimeras (Fig. 1d, e, Supp Fig. 2a). A subtle advantage is detected after the positive selection checkpoint at the CD69hiTCRβhi DP stage (Supp Fig. 2b, c, normalized ratios), but the most profound advantage for DKO thymocytes in competitive chimeras occurs at the SP stage (Fig. 1e, f, Supp Fig. 2d, e).

Strikingly, DKO cells exhibited a much larger advantage in the SP8 CD69hi stage than in the SP4 CD69hi stage (Fig. 1f). This may reflect greater escape of SP8 than SP4 DKO from deletion in the medulla. Indeed, we observe progressively reduced Nur77-GFP reporter expression during normal SP8 maturation compared to SP4 maturation (Supp Fig. 1d), which could be consistent with more stringent pruning of self-reactivity from this compartment[48,49].

### Nr4a1/3 regulate Foxp3+ populations by cell intrinsic and non-cell intrinsic mechanisms

DKO:WT chimeras reconstituted a Treg compartment of mostly WT origin – as previously reported and consistent with a requirement for the Nr4a family in Treg generation and maintenance (Fig. 1g–i, Supp Fig. 2f–h)[21–23,50,51]. However, the total thymic Treg (tTreg) compartment was expanded in DKO:WT chimeras relative to WT:WT chimeras (Supp Fig. 2f). We postulate this cell-extrinsic phenotype may be mediated by enhanced IL-2 production by self-reactive DKO thymocytes[5,52–54]. Indeed, CITEseq data reveals an IL-2/STAT5 signature overlapping with the highly signaled SP cluster ('neg sel 2') destined for deletion/diversion (Supp Fig. 1e). CD25+Foxp3- SP4 thymocytes are proposed to represent self-reactive Treg precursors[5,52]. Interestingly, the competitive advantage for SP4 DKO thymocytes was greatest in the CD25+Foxp3- precursor niche but severely abrogated among more mature CD25+Foxp3+ Treg, consistent with a partial block at this transition (Fig. 1g–i).

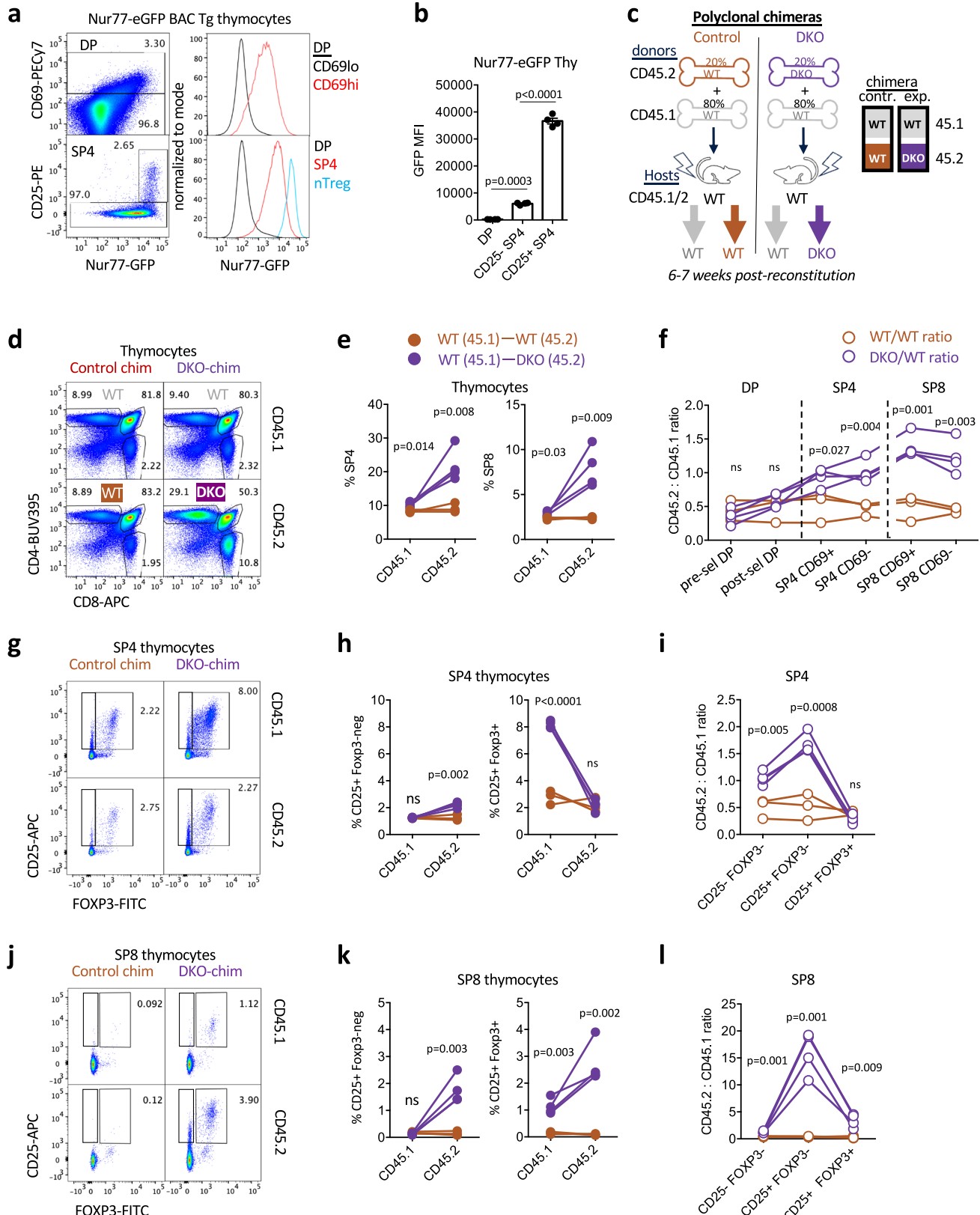

Unexpectedly, we discovered a novel CD25+ SP8 population in DKO:WT experimental chimeras that was undetectable in WT:WT control chimeras and almost exclusively populated by DKO thymocytes (Fig. 1j–l). We additionally identified a SP8 Foxp3+CD25hi population also exclusive to DKO:WT chimeras and enriched within DKO thymocytes (Fig. 1j–l, Supp Fig. 2i–k). An analogous population of Foxp3+ CD8 T cells have been described among tumor-infiltrating lymphocytes and in GVHD. Strikingly, these peripheral cells are expanded in the absence of BIM suggesting this fate may be normally censored by thymic deletion[55,56].

**Fig. 1 | Polyclonal DKO T cells exhibit a competitive advantage during thymic selection. a, b** Nur77-eGFP BAC Tg thymocytes were stained to detect thymic subsets. Lefthand panels depict representative gating to identify CD69hi post-selection DP thymocytes and CD25hi SP4 Treg. Righthand histograms depict GFP expression in gated populations. Data in a are representative of 3 independent experiments. Graph in b depicts GFP MFI quantification of 4 biological replicates +/- SEM. **c** Schematic depicts radiation chimera design. **d** Thymocytes from chimeras were stained to detect congenic markers CD45.1 and 2 as well as CD4/CD8 co-receptors. Representative plots depict thymic subsets of each donor genotype. **e** Graphs depict quantification of gating in (**d**) with lines connecting donor genotypes from individual chimeras. **f** Graph depicts ratio of CD45.2/CD45.1 donor genotypes from individual chimeras within DP and SP populations further sub-gated to define CD69hi and CD69lo compartments. **g–l** Representative plots depict SP4 (**g**) or SP8 (**j**) thymocytes of each donor genotype from both experimental and control chimeras (as above) stained to identify CD25 and Foxp3 expression. Graphs in **h, k** depict quantification of gating in **g, j** respectively with lines connecting donor genotypes from individual chimeras. Graphs in **i, l** depict ratio of CD45.2/CD45.1 donor genotypes from individual chimeras within same SP4 and SP8 subset gates. Statistical tests: **b.** one-way ANOVA with Tukey's multiple comparison test. **e, f, h, i, k, l** Unpaired t tests, parametric, assume same SD, two-sided. Graphs in **e, f, h, i, k, l** depict N = 3 WT chimeras and N = 4 DKO chimeras (biological replicates). Data in **d–i** representative of 3 independent sets of chimeras. Source data are provided as a Source Data file.

### *Nr4a1* and *Nr4a3* are required for negative selection by tissue-specific Ag

We next sought to test the hypothesis that Nr4a family members mediate negative selection by tissue-specific antigen in the medulla. To model thymocyte recognition of self-antigen expressed by mTECs, we took advantage of OT-II TCR Tg that recognizes OVA 323-339 peptide presented by Class II MHC. We generated BM chimeras with either WT hosts or hosts expressing RIPmOVA Tg in which the rat insulin promoter directs expression of a model self-antigen (membrane-tethered ovalbumin) in pancreatic beta islet cells (and in renal proximal tubular cells and testes), as well as in radioresistant mTECs[57,58].

In these chimeras, congenically marked OT-II mixed with DKO OT-II thymocytes could be tracked through development in the presence of cognate antigen expression by mTECs, enabling us to identify cell-intrinsic and Ag-dependent phenotypes (Fig. 2a). Importantly, to provide an abundant source of normal Treg, we included congenically-marked polyclonal WT donor BM (Fig. 2a). Precursor frequency of OT-II thymocytes in these chimeras is reduced to avoid overwhelming tolerance machinery[59]. Reassuringly, both WT and RIPmOVA hosts reconstitute a normocellular thymus and Treg compartment (Supp Fig. 3a, b).

We found that all three *Nr4a* transcripts were induced in semi-mature CD69hi OT-II (Vα2+ Vβ5+) SP4 thymocytes from negatively-selecting RIPmOVA hosts, although absolute abundance of *Nr4a1* far exceeded that of *Nr4a2* and *Nr4a3* (Fig. 2b). OT-II thymocytes of either genotype represent a small and comparable proportion of the DP gate, yet they outcompete polyclonal thymocytes into the SP4 compartment of WT hosts (Fig. 2c, Supp Fig. 3c–e). This may reflect more efficient positive selection of OT-II and/or reduced negative selection relative to polyclonal cells. By contrast to OT-II thymocytes, DKO OT-II cells exhibit a marked competitive advantage into the SP4 compartment and are not deleted in response to RIPmOVA expression (Fig. 2c, Supp Fig. 3c–f).

Importantly, only a very small fraction of OT-II is diverted into the Treg compartment (Fig. 2d–f, Supp Fig. 3c), consistent with prior observations that this TCR Tg model favors deletion rather than Treg induction[58]. Instead, Treg in this chimera model predominantly originate from polyclonal WT donor BM irrespective of host genotype.

### *Nr4a1* and *Nr4a3* mediate negative selection of CD69hi semi-mature SP thymocytes

We next sought to define the stage at which OT-II and DKO OT-II development diverge in these chimeras. CD69 is upregulated at the positive selection checkpoint in DP thymocytes, remains high on semi-mature SP thymocytes and is then downregulated in mature SP thymocytes prior to thymic export (Supp Fig. 1e)[60]. In RIPmOVA hosts, counter-selection of OT-II thymocytes occurs between the SP4 CD69hi and the mature SP4 CD69lo stage (Fig. 2g). This is consistent with CITEseq data showing that *Ccr7*hi SP cluster destined for negative selection in the medulla express high levels of CD69 protein on their surface (Supp Fig. 2e). Indeed, almost no OT-II mature CD69lo SP4 thymocytes survive in RIPmOVA hosts (Fig. 2h, i, Supp Fig. 3f). By contrast, DKO OT-II thymocytes evade counter-selection in RIPmOVA

hosts and populate the mature CD69lo SP4 compartment. Importantly, analysis of spleen reveals abundant DKO OT-II but near-complete loss of OT-II CD4 T cells in RIPmOVA hosts (Fig. 2j). Advantage of DKO OT-II thymocytes relative to OT-II even in WT hosts may reflect differential response to endogenous pMHC (Fig. 2c, j, Supp Fig. 3c–f). Importantly, this advantage is dramatically enhanced in RIPmOVA hosts, confirming the dependence of the phenotype on mTEC expression of antigen.

Finally, OT-II SP thymocytes in RIPmOVA hosts exhibit marked downregulation of surface Vα2+ Vβ5+ TCR levels, while DKO OT-II thymocytes retain high surface TCR expression (Fig. 2k, l, Supp Fig. 3g, h) This model enabled us to capture CD69hi SP4 OT-II cells before they are lost to deletion and without diversion towards Treg fate, and therefore served as a unique platform to define the mechanism by which the Nr4a family regulates thymic deletion in the medulla via transcriptional profiling.

### Nr4a3 is completely dispensable for deletion of OTII thymocytes in RIPmOVA hosts

Importantly, analogous chimeras generated with *Nr4a3*-/- OT-II donors in lieu of DKO OT-II BM reveal no substantial escape from deletion, preserved Treg induction, and minimal impact on surface TCR downregulation in RIPmOVA hosts (Supp Fig. 3i–l). This data – together with previously characterized *Nr4a1*-/- OTII/RIPmOVA model with only a subtle impact on deletion[17] – confirms significant redundancy between Nr4a1 and Nr4a3 in medullary thymic tolerance.

### *Nr4a1* and *Nr4a3* are required for thymic transcriptional upregulation of *Bcl2l11*/BIM by self-antigen recognition

The pro-apoptotic BH3-only domain Bcl2 family member BIM is among the few proteins with a well-defined functional role during thymic negative selection[10,61]. BIM is transcriptionally upregulated by TCR signaling and operates by triggering mitochondrial cytochrome c release and caspase activation[61,62]. T cells from *Bcl2l11*/BIM-deficient mice escape from negative selection by superantigen, peptide injection, and in the HY model of ubiquitous antigen expression[10,19]. In addition, BIM-deficient SP4 thymocytes escape deletion in the InsHEL model in which HEL antigen is expressed in mTECs, similar to RIPmOVA[42,63]. CITEseq data revealed that *Bcl2l11* transcript is highly expressed among deleting thymocytes clusters and correlated with expression patterns of *Nr4a1* and *Nr4a3* (Fig. 3a, Supp Fig. 1e). We found that BIM protein expression is upregulated in post-selection DP and in SP thymocytes in proportion to Nur77-GFP reporter expression (Fig. 3b, c, Supp Fig. 4a). Consistent with prior reports, post-selection DP and SP thymocytes are expanded in the absence of BIM, and Nur77-eGFP reporter expression among those populations is markedly increased corresponding to rescue of strongly signaled thymocytes from deletion (Fig. 3d, e)[35,64]. Taken together, these results suggest that the highest Nr4a1-expressing thymocytes also express BIM levels high enough to bind and sequester anti-apoptotic Bcl2 family members in those cells, thereby tipping the balance in favor of apoptosis.

To test whether Nr4a family regulates *Bcl2l11* transcription, we sorted semimature CD69hi SP4 Vα2+ Vβ5+ OT-II thymocytes of each

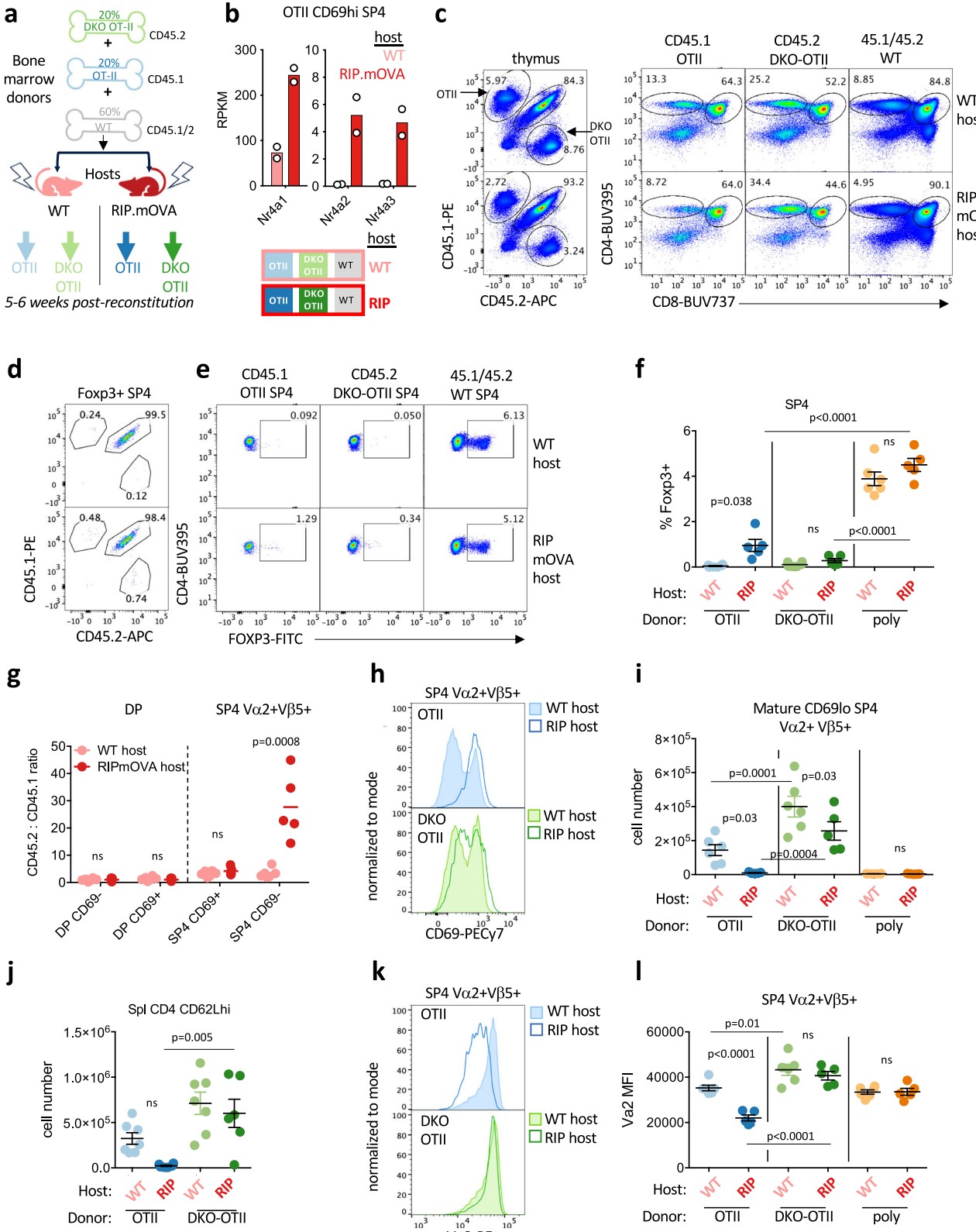

donor genotype from either WT or RIPmOVA hosts (as in schematic Fig. 2a) and compared their transcriptomes. RNAseq analysis of OT-II and DKO OT-II semimature thymocytes shows comparable and robust induction of *Gadd45b* (Fig. 3f, Supplementary Data 1a), a previously identified transcriptional marker of negative selection that is also correlated with *Bcl2l11* transcript in clusters of deleting thymocytes in CITEseq atlas (Supp Fig. 4b)[65,66]. This indicates that both OT-II and DKO

OT-II sense self-antigen in RIPmOVA hosts, and we successfully captured OT-II thymocytes prior to their deletion. By contrast, *Bcl2l11* transcript and BIM protein was induced in OT-II but not in DKO OT-II from RIPmOVA hosts (Fig. 3g–i). Importantly, BIM expression was normally induced in *Nr4a3-/-* SKO OT-II thymocytes from RIPmOVA host chimeras - consistent with intact deletion of those cells - demonstrating redundancy between Nr4a1 and Nr4a3 (Supp Fig. 4c, d).

**Fig. 2 | Nr4a1 and Nr4a3 mediate negative selection of CD69hi SP thymocytes. a** Schematic depicts radiation chimera design. **b** Graph depicts rpkm of *Nr4a1-3* transcripts in CD69hi SP4 va2+vb5+ OT-II thymocytes sorted from either WT or RIPmOVA host chimeras in biological replicate (schematic above in a) and subjected to RNAseq (Supplementary Data 1a depicts p value and fdr for WT vs RIP-mOVA hosts, GSE235101). **c** representative plots depict thymocytes from chimeras in (**a**) harvested 5-6 weeks post irradiation and BM transfer and stained to detect (left) donor genotype and (right) thymic subsets among each donor genotype from WT host (top) or RIPmOVA host (bottom). **d**–**f** thymocytes from chimeras harvested as above were stained to detect intracellular Foxp3 expression in addition to congenic markers and co-receptors. Representative plots depict (**d**) Foxp3 + SP4 thymocytes stained to detect donor genotype and (**e**) SP4 thymocytes of each donor genotype (as gated in c) stained to detect Foxp3+ cells. Graph (**f**) depicts quantification of gates in (**e**) +/- SEM. **g** Graph depicts ratio +/- SEM of CD45.2/ CD45.1 donor genotypes from individual chimeras within DP and SP populations further sub-gated to define CD69hi and CD69lo compartments. **h, i** Representative overlaid histograms depict CD69 expression among SP4 Vβ5 + Vα2+ thymocytes of each OTII donor genotype from WT hosts and RIPmOVA hosts as color-coded in (**a**). Absolute number (**i**) of CD69lo cells (based on bimodal CD69 distribution in top panel of h) are quantified +/- SEM. **j** Graph depicts naïve CD4 splenic T cell count +/- SEM of each donor genotype (OT-II or DKO-OT-II) from WT or RIPmOVA host chimeras (see schematic above in a). **k, l** Representative overlaid histograms depict surface expression of TCR Vα2 among Vβ5 + Vα2 + SP4 thymocytes of each OT-II donor genotype from WT hosts and RIPmOVA hosts as color-coded in (**a**). Graph depicts quantification of Vα2 MFI in this population +/- SEM. Statistical tests: **f, j, l** one way ANOVA with Tukey's multiple hypothesis test. **g** multiple unpaired t-tests with Holm-Sidak correction for multiple comparisons, assume SD across comparisons, two sided. **i** one way ANOVA with pre-specified comparisons corrected for by Sidak test. Graphs in this figure **f, g, i, j, l** depict N = 6 WT host and N = 5 RIPmOVA host chimeras – biological replicates. Data in this figure are representative of 2 independent sets of chimeras. Source data are provided as a Source Data file.

As in the RIPmOVA model, polyclonal DKO thymocytes also exhibited a large reduction in BIM expression, predominantly in SP thymocytes (Fig. 3j, k). Importantly, in vitro induction of BIM with anti-CD3ε stimulation was almost entirely abolished in DKO SP4 thymocytes, suggesting Nr4a1 and Nr4a3 are required for Ag-dependent BIM induction to mediate deletion (Supp Fig. 4e, f).

Because BIM and other BH3-only family members function through stoichiometric sequestration of anti-apoptotic partners, we next assessed transcript abundance of Bcl2 family "executioners" Bax and Bak, their activators (BH3-only pro-apoptotic family members), and their anti-apoptotic inhibitors in our dataset (Supp Fig. 4g, Supplementary Data 1a). Among pro-apoptotic members, *Bax* is constitutively expressed and *Bcl2l11*/BIM is induced by RIPmOVA in an Nr4a-dependent manner, but all others are minimally expressed including *Bbc3*/PUMA (despite its genetic implication in this pathway) and *Pmaip1*/NOXA[67]. Of the anti-apoptotic Bcl-2 family members, *Mcl* and *Bcl2* are the most abundant in our dataset but were not induced by antigen stimulation in SP4. The pro-apoptotic members *Bcl2a1* and *Bcl2b1* were induced by antigen and correlated to negatively selecting clusters in CITEseq but were Nr4a-independent (Supp Fig. 4b, g). Of note, *Bcl2* levels are slightly lower in DKO OT-II than OT-II from RIP-mOVA hosts (log2FC = −1.172) but much less reduced than *Bcl2l11* (log2FC =−2.533) (Supplementary Data 1a, Supp Fig. 4g). This suggests that the stoichiometric balance between pro- and anti-apoptotic family members in DKO thymocytes is shifted to favor survival.

### Conserved Nr4a motifs in a thymic enhancer of Bcl2l11 transcription

We next sought to determine how Nr4a factors induce *Bcl2l11* transcription. We searched publicly available ATACseq data (Immgen.org) to identify well-defined Nr4a consensus DNA binding motifs in putative cis-regulatory elements near *Bcl2l11*. Peaks at regions of open chromatin (OCR) corresponding to the TSS (transcriptional start site) of *Bcl2l11* did not contain Nr4a consensus motifs. However, we identified a limited number of OCRs within several hundred kb of *Bcl2l11* TSS with highly conserved Nr4a motifs (Supplementary Data 1b). Among these, a pair of peaks within approximately 5 kb of one another and 200 kb upstream of Bcl2l11 TSS (located within intron 9 of the adjacent gene *Acoxl*) showed uniquely high accessibility in thymocytes and T cell lineage populations but were undetectable in B cells (Fig. 3l, Supp Fig. 4h). Moreover, these corresponded to a putative enhancer (termed $E^{BAB}$) identified on the basis of independent genomic data sets (H3K27ac CHIPseq and DHS)[68]. CRISPR-Cas9 deletion of this $E^{BAB}$ cis-regulatory element previously revealed that it is required for thymic induction of *Bcl2l11* transcript and negative selection[68]. We have therefore termed these two OCRs $E^{BAB1}$ and $E^{BAB2}$ and propose that TCR-induced Nr4a family members bind directly to these enhancers to induce *Bcl2l11*

transcription and drive negative selection. Indeed, we identified an *Nr4a1*/Nur77 ChIPseq peak at the Nr4a motif in $E^{BAB1}$ (GSE102393, Supp Fig. 4h)[69]. Strikingly, we found that the adjacent *Acoxl* gene (within which $E^{BAB}$ sits) similarly exhibited impaired induction in DKO thymocytes suggesting it may be co-regulated with *Bcl2l11* by the Nr4a family via this cis regulatory element (Fig. 3m).

### Self-antigen encounter in the medulla induces a transcriptional program associated with negative selection

We next took advantage of our model to define the global transcriptional response of SP thymocytes encountering model self-antigen expressed in mTECs (chimera schematic Fig. 2a). 350 transcripts are differentially expressed in WT OT-II semimature SP4 thymocytes sorted from WT and RIPmOVA hosts (|log2FC|>1, FDR < 0.05; Fig. 4a; Supplementary Data 1a). Of these, the majority are upregulated, and only a small number of genes are downregulated. This program is highly conserved (82/350 transcripts) across the related model system in which model antigen hen egg lysozyme (HEL) is expressed under the control of the insulin promoter (InsHEL) together with the 3A9 HEL-specific TCR Tg, but minimally shared with the HY-CD4 model in which ubiquitous self-antigen deletes cognate TCR Tg cells early at the DP stage in the cortex (Supp Fig. 5a, b, Supplementary Data 2a, b)[39,66]. Notably, *Nr4a1, Nr4a3*, and *Bcl2l11* are among the limited set of shared genes induced in each of these models.

We next compared antigen-induced transcripts in our bulk RNA-seq data with the set of transcripts enriched in *Bcl2l11*+ CITEseq clusters fated for negative selection (Fig. 3a). As noted earlier, two deleting clusters are identified, one corresponding to CCR7- CD69+ DP thymocytes likely in the cortex, and the second corresponding to CCR7+ CD69hi semimature SP thymocytes likely encountering Ag in the medulla (Supp Fig. 1e). Correlation between our OT-II/RIPmOVA data and the medullary deleting cluster is remarkably high, again suggesting a highly conserved transcriptional program is triggered by self-antigen recognition in the medulla (Fig. 4b, Supplementary Data 2c). By contrast, the transcriptional programs in cortical DP thymocytes and in medullary SP thymocytes undergoing negative selection diverge; only a limited number of DEG are shared between TCR Tg models and both deleting clusters, including the *Nr4a* genes and *Bcl2l11*/BIM (Supp Fig. 5c, d, Supplementary Data 2d).

The cluster of medullary thymocytes undergoing negative selection are adjacent to tTreg on the CITEseq UMAP, highlighting their transcriptional similarity to one another (Supp Fig. 5e–g, Supplementary Data 2d). While *Bcl2l11* and *Gadd45b* are higher in cells fated for deletion than in Treg, several transcripts upregulated in response to RIPmOVA mark both of these populations, such as the transcription factor *Ikzf2*/HELIOS, the co-stimulatory receptors *Tnfrsf4*/OX40, *Tnfrsf/*41BB, as well as *Nr4a1* and *Nr4a3* themselves (Supp Fig. 5e, f). This

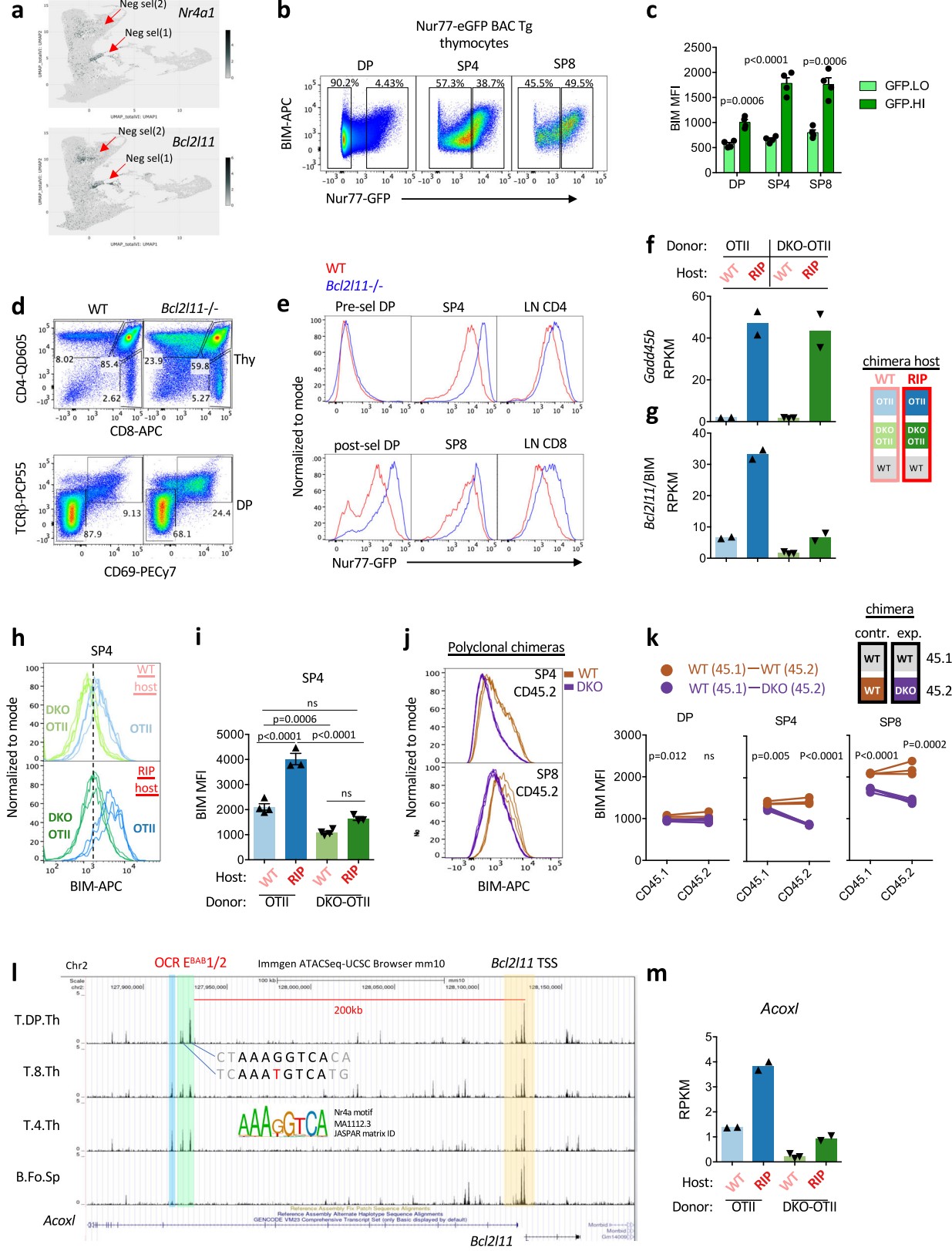

is consistent with establishment of a common precursor pool of self-reactive thymocytes[52]. Importantly, our sorted OT-II samples express virtually no detectable *Foxp3* transcript and also have low *Il2ra* levels suggesting that profiled thymocytes were captured **_before_** commitment to either deletion or diversion and are distinct from CD25hi pre-Treg[52] (Supp Fig. 5g).

### Nr4a1 and Nr4a3 are required to enforce a broad SP thymocyte transcriptional program in response to self-antigen recognition in the medulla

We next sought to determine how the large and conserved transcriptional program induced by antigen recognition in the thymus was regulated by the Nr4a family. Principal component analysis revealed

**Fig. 3 | Nr4a1 and Nr4a3 are required for upregulation of Bcl2l11/BIM in SP thymocytes by self-Ag recognition. a** *Nr4a1* and *Bcl2l11* transcript UMAP data from publicly available VISION thymus CITEseq interface. Red arrows indicate two clusters of negatively selecting cells, early in cortical DP and late in medullary SP. See also Supp Fig. 1e for additional annotation. **b, c** Nur77-eGFP BAC Tg thymocytes were stained to detect thymic subsets and intra-cellular BIM protein. Representative plots depict correlation between markers and show GFPlo/hi gates. Graph (**c**) depicts MFI of BIM among GFPhi and GFPlo populations as gated in (**b**) +/− SEM from 4 biological replicates. **d, e** Nur77-eGFP BAC Tg was crossed onto the *Bcl2l11*−/− background. Thymocytes from *Bcl2l11* +/+ and −/− reporter mice were stained to detect SP subsets (**d**, top panels) and gating for pre-selection (CD69lo TCRβlo) and post-selection (CD69hiTCRβhi) DP thymocytes (**d**, bottom panels). Histograms depict reporter GFP expression in overlaid genotypes gated as in (**d**) along with peripheral CD4, CD8 LN T cells. Data are representative of 3 biological replicates. **f, g** Graphs depict RPKM for *Gadd45b* (**f**) and *Bcl2l11* (**g**) in CD69hi Va2 + SP4 thymocytes of either WT-OT-II or DKO-OT-II donor genotypes from chimeras depicted in 3a. RNAseq was performed on biological replicates (Supplementary Data 1a depicts p value and fdr for all pairwise comparisons across sample types, GSE235101). **h, i** Overlaid histograms depict intracellular BIM protein expression in SP4 OT-II and DKO-OT-II thymocytes from N = 4 WT host and N = 3 RIPmOVA host chimeras – biological replicates (as schematized in 3a). Graph (i) depicts BIM MFI +/− SEM from these samples. **j, k** Overlaid histograms depict intracellular BIM protein expression in CD45.2 donor genotypes in SP4 (top) and SP8 (bottom) thymocytes from N = 3 control WT:WT chimeras and N = 4 experimental DKO:WT polyclonal chimeras – biological replicates (as schematized in 1c). Graphs (**k**) depict BIM MFI across thymic subsets in both CD45.2 and CD45.1 donor genotypes from both types of chimeras. Lines connect samples from the same chimera. **l** UCSC browser tracks depict ATACseq peaks from public immgen.org data across thymic subsets and Follicular splenic B cells for reference at the *Acoxl/Bcl2l11* locus. 200 kb upstream of the TSS of *Bcl2l11* are green-tinted ATAC peaks with consensus Nr4a sites that correspond to EBAB enhancer identified in Hojo et al. (PMID 31197149). Blue tint identifies an adjacent peak with striking cell type specificity but only an imperfect Nr4a site. See also Supplementary Data 1b. **m** Graph depicts RPKM for *Acoxl* as in **f, g**, statistical analyses in Supplementary Data-Table 1a. Statistical tests: **c, k** unpaired parametric two-tailed t-test, assume equal SD. **i** one way ANOVA with Tukey's multiple hypothesis test. Source data are provided as a Source Data file.

that DKO OT-II and WT OT-II thymocytes diverged markedly in RIPmOVA (but not WT) hosts, suggesting the Nr4a family plays a selective role in self-reactive thymocytes (Supp Fig. 6a, b, Fig. 4c). We identified genes that were differentially expressed in Nr4a-sufficient and Nr4a-deficient OT-II thymocytes from RIPmOVA hosts (Fig. 4c, d). To visualize this program across both donor and host, we identified transcripts differentially expressed between any pairwise comparison of the four sorted thymocytes populations (schematic Fig. 2a, fdr <0.05, |log2FC| ≥ 1), and generated a heatmap focused on genes that are upregulated by antigen recognition (log2FC > 0 OT-II-RIP/OT-II-WT, N = 551) (Fig. 4e, Supplementary Data 1a). A set of RIPmOVA-induced transcripts is shared between WT OT-II and DKO OT-II thymocytes (Fig. 4a, d, e, Supp Fig. 6c). These include *Gadd45b* (as noted earlier), *Ikzf2*/HELIOS as well as inhibitory receptors *Pdcd1*/PD1, *Cd200*, co-stimulatory receptor *Tnfrsf4*/OX40, and the ubiquitin ligases *Cblb* and *Tnfaip3*/A20. These data confirm that there is no global defect in antigen sensing or TCR signaling in DKO thymocytes.

Notably, over half of the RIPmOVA-induced transcriptome is almost entirely lost in the absence of Nr4a1/3 (Fig. 4e). This was similar to a broad - but not global - transcriptional defect previously described in the autoimmune-prone NOD genetic background that was also associated with impaired *Bcl2l11* induction and defective negative selection (Supp Fig. 6d, Supplementary Data 2a)[40,66]. The Nr4a-dependent program we identify encompasses Treg-associated transcripts (e.g. *Foxp3* itself despite minimal reads, *Ikzf4, Ahr, Lrrc32*/GARP), and negative selection-associated transcripts *Lad1, Arhgap20*, as well as *Bcl2l11*/BIM itself (Fig. 4c, d, e). We also identified Nr4a-dependent transcripts *Rln3, Eno3, Itih5* and *Tnfrsf9*/41BB previously reported in thymocytes and peripheral T cells, including *Acvrl* which is adjacent to *Nr4a1* in the mouse genome (Fig. 4c–e, Supp Fig. 6b, e, f, Supp Fig. 7a, b, Supplementary Data-Tables 2e–h)[17,50]. We also confirmed Nr4a-dependence of *Ndg2/Chchd10* – a Nur77-induced transcript with pro-apoptotic functions identified by Winoto and colleagues, but we find that other putative Nur77 targets reported in that study are Nr4a-independent (*Fasl, Pdcd1*/PD1, TRAIL/*Tnfsf10*, *Ctla4*, and Ndg1/*Khdc1*)[25].

**Deletion-associated transcriptome correlates with Nur77/Nr4a1-GFP reporter expression in the normal SP4 thymocyte repertoire**

We showed that "self-reactive" OT-II SP4 thymocytes destined for deletion in RIPmOVA hosts express a conserved and partially Nr4a-dependent transcriptional program. We next sought to determine whether this program correlated with Ag-recognition and Nr4a expression in the normal thymic repertoire. To that end, we analyzed the transcriptome of semimature SP4 thymocytes (CD25-CD69hiCD24lo) sorted from Nur77/Nr4a1-GFP reporter mice according to GFP expression (Fig. 5a, b, Supp Fig. 8a). Sorted populations faithfully captured low, medium, and high levels of *Gfp* and *Nr4a* transcripts (Fig. 5c). We identified DEG between GFPlo and GFPhi thymocytes (Supplementary Data 3a, Fig. 5d, e, |Log2FC| ≥ 1 and padj <0.05). The majority of DEG were upregulated rather than down-regulated in GFPhi thymocytes, while GFPmed thymocytes largely exhibited intermediate gene expression between GFPlo and GFPhi populations (Fig. 5b, d, Supplementary Data 3a). This suggests that we captured a gradient of self-reactivity and Ag-dependent transcription programs. We next compared these DEG to the transcriptome induced in OT-II thymocytes by RIPmOVA recognition. Half of the OT-II/RIPmOVA transcriptome was shared with GFP-HI SP4, constituting a common program (Fig. 5f, g, Supplementary Data 3b). Of note, over a third of these shared DEG required Nr4a1/3 for their induction, implying Nr4a dose-dependent control of their expression (Supplementary Data 3a, b).

**Self-antigen encounter in the medulla induces an anergy-like transcriptional program that is partially dependent on Nr4a1/3**

We next sought to understand the functional significance of this Ag-dependent thymic transcriptional program identified in both the OTII-RIPmOVA model and in the natural thymic repertoire. When we compare these genes to TCR-dependent transcripts in peripheral T cells, we noted that only a minority represent rapidly induced primary/early response genes, while the bulk of the self-antigen-induced transcriptome in medullary SP thymocytes strongly resembles that of naturally occurring anergic T cells (Fig. 5h, Supplementary Data 4a). Importantly, this is confirmed by formal gene set enrichment analysis (Fig. 5i, j).

Anergic gene expression is induced when T cells encounter signal 1 (TCR) in the absence of signal 2 (CD28 co-stimulation). Consistent with this, RIPmOVA-induced thymic transcripts overlap with a CD4 T cell tolerance program induced in vitro by TCR stimulation without co-stimulation (Supplementary Data 4b)[70]. Signal 1 in the absence of Signal 2 produces NFAT nuclear translocation in the absence of AP-1 and NF-kB[71]. Anjana Rao and colleagues showed that a constitutively active form of NFAT that cannot cooperate with AP-1 (CA-NFAT-RIT) drives an anergic transcriptional program[72]. Here again we identified a set of NFAT-responsive genes that overlapped with our data set (Supplementary Data 4c) and account for some of the anergy and exhaustion programs discovered in peripheral T cells such as *Nr4a2/3, Tnfrsf4* and 9, as well as *Cblb* and *Bhlhe40* among others. Similarly, markers of bona fide CD8 T cell exhaustion are shared with OT-II thymocytes from RIPmOVA hosts (Supplementary Data 4d)[73].

Importantly, many anergy-associated genes induced by RIPmOVA required Nr4a for their induction (Fig. 4e). We discovered

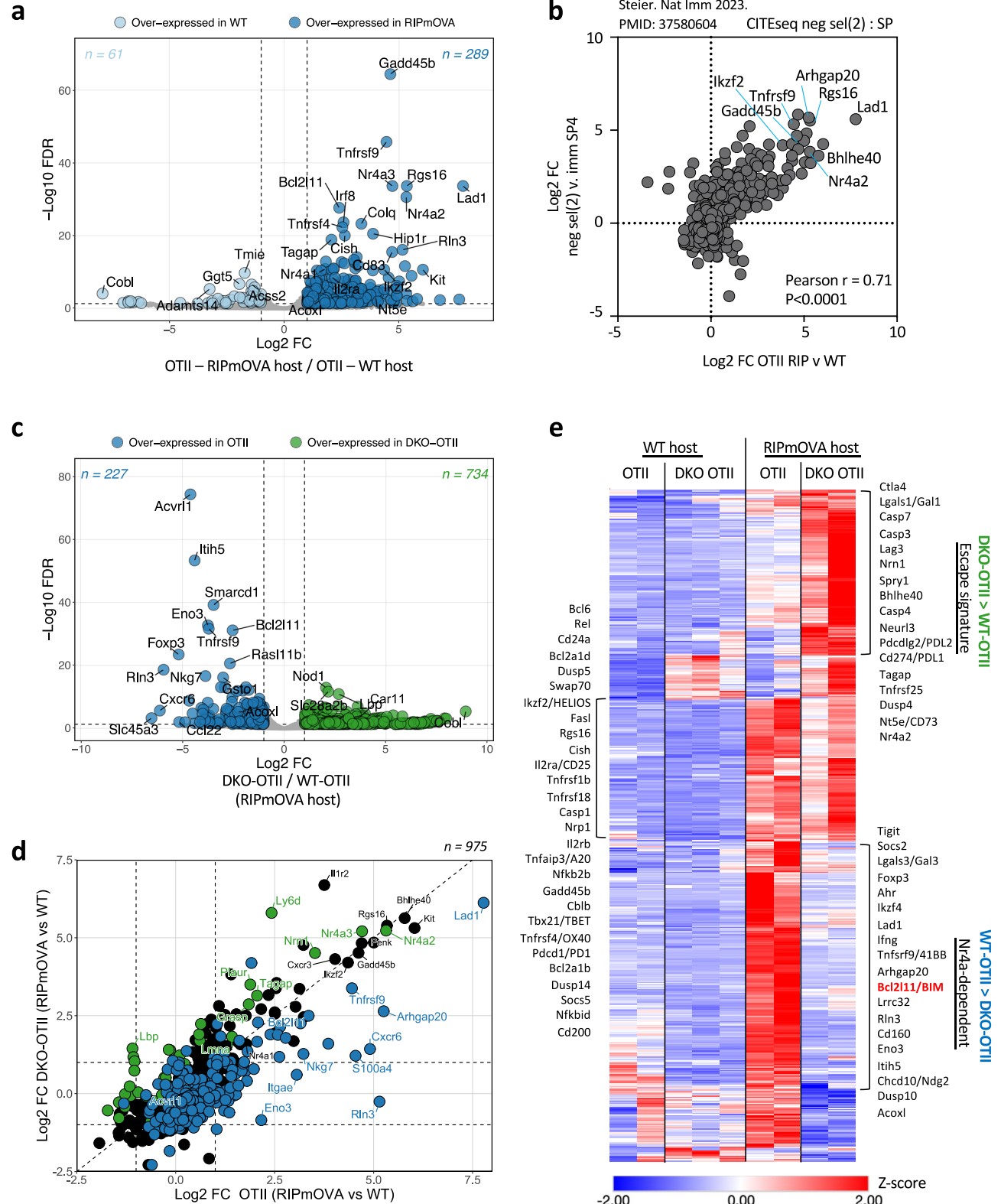

negative regulators of TCR and cytokine signaling in this module, including cell surface molecules *Tigit, Cd160, Lgals3*/Galectin-3, as well as intracellular suppressor of cytokine signaling *Socs2* and MAPK PTPase *Dusp10*. Reducing stringency of filtering (p < 0.01) to increase discovery of RIPmOVA-induced genes also identifies *Cd5, Dusp5, Dusp1, Tox2*, and *Cish* as negative regulators modulated by the Nr4a family (Supplementary Data 1a, GSE235101).

## Nur77-eGFP reporter marks population of naïve CD4 T cells with an "anergic" phenotype that are enriched for recent thymic emigrants

We recently described a population of naturally occurring Nur77/Nr4a1-eGFP-hi naïve CD4 T cells (CD62Lhi CD44lo CD25- Foxp3-) characterized by low Ly6C expression (Fig. 6a). By contrast to other naïve CD4 T cells, these Nur77-GFPhi Ly6Clo cells (previously

**Fig. 4 | *Nr4a1* and *Nr4a3* mediate a broad transcriptional program induced by antigen recognition in the thymus. a** Bulk RNA sequencing was performed on CD69hi Vα2 + SP4 thymocytes of WT OT-II and DKO OT-II donor genotypes sorted from both WT and RIPmOVA hosts, as depicted in 2a. Differential gene expression analysis was undertaken for all pairwise comparisons; |log2FC | ≥ 1 and fdr <0.05 was considered significant. See also Supplementary Data 1a, GSE235101. Volcano plot depicts comparison between WT OT-II thymocytes sorted from RIPmOVA vs WT hosts. Colored points identify genes that meet significance threshold above. **b.** FC FC plot compares gene expression changes between WT OT-II thymocytes from RIPmOVA hosts vs WT hosts (see also panel a, above) with gene expression changes between "neg sel 2" cluster vs "immature SP4" cluster as determined by single cell methods and VISION analysis (Steier et al., 2023). Pearson correlation coefficient is quantified. See also Supplementary Data 2c. **c** As in (**a**) except volcano plot depicts DEG (colored points) in DKO OT-II vs WT OT-II thymocytes sorted from RIPmOVA hosts. **d** FC FC plot compares gene expression changes in WT OT-II thymocytes

(RIPmOVA compared to WT hosts) to those in DKO OT-II thymocytes (RIPmOVA compared to WT hosts). Genes with RPKM > 5 for all samples in at least one experimental group and FDR < 0.05 in any pairwise comparison (see a) are plotted. Genes with log2FC ≥ 1 and FDR < 0.05 in DKO OT-II vs WT OT-II (in RIPmOVA hosts) are colored green; genes with log2FC ≤ 1 and FDR < 0.05 in DKO OT-II vs WT OT-II (in RIPmOVA hosts) are colored blue. Select gene names are included. **e** Heatmap depicts normalized gene expression changes (z-score) across WT OT-II and DKO OT-II biologic replicate samples sorted from WT and RIPmOVA hosts. Shown are genes with |log2FC | ≥ 1 and fdr <0.05 in any of the four pairwise comparisons (described in a) that are also upregulated by thymic antigen encounter (log2FC > 0 in WT OT-II thymocytes from RIPmOVA compared to WT hosts; N = 551; Supplementary Data 1a). Annotated modules depict selected genes that are upregulated, downregulated, or comparably expressed in DKO OT-II compared to WT OT-II thymocytes sorted from RIPmOVA hosts.

termed population "D") express inhibitory molecules, exhibit impaired response to TCR stimulation, and preferentially differentiate into Treg[74,75]. In the thymus, SP4 cells predominantly express low levels of Ly6C[76] and there is relative enrichment for Nur77-GFPhi cells that have a phenotype that resembles a population "D" phenotype in peripheral CD4 T cells (Fig. 6b). This finding raised the possibility that recent thymic emigrants (RTE) could contribute to the composition of this previously described peripheral population "D".

To determine the relationship between RTE and population "D" cells, we used the RAG-GFP reporter, which faithfully identifies bona fide RTE[77]. To identify population "D"-like cells without the Nur77-eGFP reporter, we used CD5 in place of Nur77-eGFP, because the intensities of both markers are positively correlated[75]. Using the combination of CD5 and Ly6C, we identified populations A'-D' where Nur77-GFP MFI increased from A' to D', as expected (Fig. 6c). In RAG-GFP reporter animals, population D' was enriched for cells with the highest GFP MFI, indicating an enrichment for RTE (Fig. 6c, d).

In addition, we adoptively transferred Nur77/Nr4a1-GFP reporter thymocytes into WT hosts and tracked their phenotype in the naïve CD4 compartment over time (Fig. 6e). Indeed, most adoptively transferred thymocytes initially exhibit a Pop D phenotype (Ly6Clo, Nur77/Nr4a1-GFP hi). The Pop D proportion of transferred cells progressively waned over 10 days – likely due to both attrition and a changing phenotype – approaching a steady state distribution of naïve CD4 T cells across Pop A-D (Fig. 6f, Supp Fig. 8b, c)[74]. These data collectively support the conclusion that Pop D is highly enriched for bona fide RTE, and this feature correlates with their tolerogenic properties[74,75].

### CD4+ recent thymic emigrants exhibit epigenetic imprint corresponding to thymic Ag encounter

ATACseq analysis revealed a marked increase in chromatin accessibility across a set of approximately 2000 genes in Nur77-eGFPhi population D relative to other naïve CD4 T cells and this signature correlated with RTE enrichment, suggesting that it might reflect recent thymic Ag recognition (GSE206074, Fig. 6g)[74]. We found that DEG upregulated in Nur77-GFPhi thymocytes overlapped markedly with DAR in Pop D cells (Fig. 6g, Supplementary Data 4e). These include genes associated with anergy and exhaustion. Similarly, DEG upregulated in OT-II thymocytes from RIPmOVA hosts were strikingly enriched in Pop D DAR; almost half of all RIP-induced genes exhibited greater chromatin accessibility in Pop D/RTEs (Fig. 6h, Supplementary Data 4f. Moreover, among the overlapping genes, nearly half require Nr4a1/3 for their induction (41/114). Together, these findings led us to postulate recent antigen-encounter in the thymus left an epigenetic imprint (partially mediated by the Nr4a family) on RTE in Pop D that confers an "anergy"-like state in the periphery. Importantly, this is a feature of the natural thymic and

naïve CD4 T cell repertoire (as marked by Nur77-eGFP reporter expression), and not merely restricted to OT-II-RIPmOVA thymocytes destined for deletion.

### Nr4a-deficient thymocytes exhibit a transcriptional escape signature

In addition to identifying genes that depend on the Nr4a family for their induction in self-reactive OTII thymocytes from RIPmOVA hosts, we observed a module of antigen-induced transcripts that were over-expressed in the absence of Nr4a1/3 (Fig. 4e). We hypothesize that these genes largely represent an "escape signature" in DKO SP4 thymocytes that were destined for – but evaded –- deletion; over-induced genes include those that are upregulated by strong TCR signaling but do not require Nr4a1/3 for their induction.

Indeed, a similar "escape signature" was described among thymocytes evading negative selection in $E^{BAB}$ KO mice with a deletion of the thymic *Bcl2l11*/BIM enhancer (Supp Fig. 8d, e, Supplementary Data 4g)[68]. Strikingly, some (N = 33) of this TCR-dependent escape signature was Nr4a-dependent including *Arhgap20, Lad1, Tnfrsf9, Eno3* among others. This is consistent with our model that the Nr4a family is upstream of the $E^{BAB}$ enhancer to induce *Bcl2l11*.

By contrast, the escape signature in DKO thymocytes represent TCR-dependent genes that do not require Nr4a1/3 for their induction (Supp Fig. 8e, schematic model), and includes many anergy-associated inhibitory cell surface molecules: *Ctla4, Lgals1*/Galectin-1, genes encoding both PDL1 (*Cd274*) and PDL2 (*Pdcdlg2*), and *Lag3* as well as the anergy-associated marker *Nt5e*/CD73 which is expressed on naturally-occurring anergic populations (Fig. 4e)[78]. Additional negative regulators and anergy-associated genes in this module included *Dusp4*, the E3 ubiquitin ligase *Neurl3*, the MAPK negative regulator *Spry1*, the NFAT-induced TF *Bhlhe40*, as well as caspases *Casp3/4/7* (Fig. 4e).

### HELIOS and CD73 identify Nr4a-deficient thymocytes that escape antigen-dependent deletion

We next sought to identify DKO thymocytes fated for – but rescued from – negative selection by flow cytometry. We focused on RIPmOVA-induced transcripts that do not require Nr4a1/3 for their induction and therefore could mark strongly signaled thymocytes escaping deletion (Fig. 7a-d). HELIOS is encoded by *Ikzf2*, is highly expressed in Treg, and was previously identified as a marker of Foxp3-negative thymocytes escaping negative selection[42]. Indeed, as noted earlier, *Ikzf2* transcript is upregulated by RIPmOVA in our bulk data set, and it is also highly expressed in both negatively selecting clusters in CITEseq data (Fig. 7a, b). HELIOS protein expression is high (comparable to Treg) in OT-II thymocytes from RIPmOVA hosts but virtually undetectable in WT hosts (Fig. 7e, f). Flow cytometric analysis of protein expression reveals a larger fraction and much larger total number of DKO OT-II SP4 thymocytes are HELIOS positive (Fig. 7e–g). We propose these

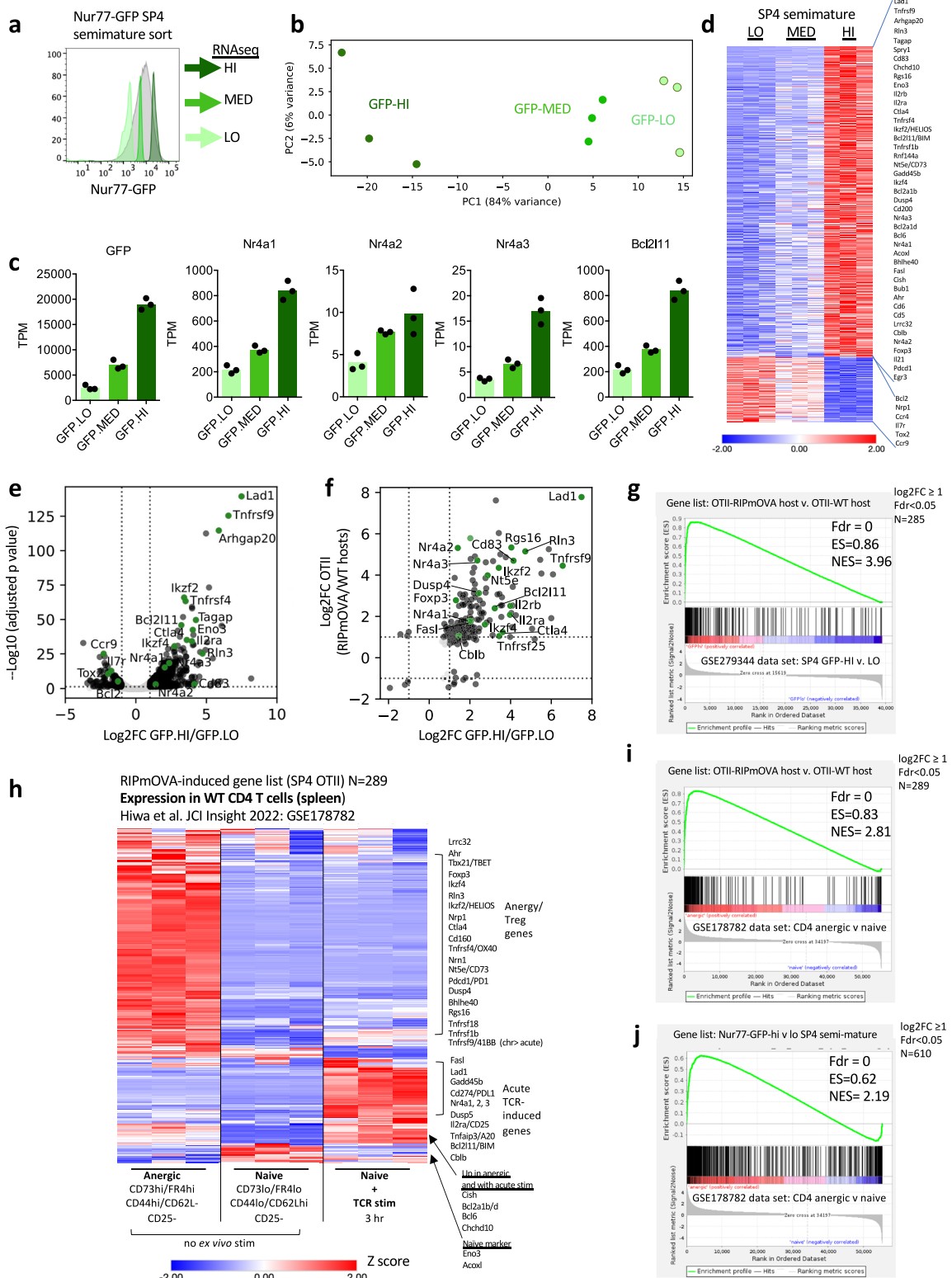

Foxp3-negative cells represent DKO thymocytes that have escaped negative selection.

Among transcripts enriched in DKO OT-II cells relative to WT OT-II from RIPmOVA hosts (so-called "escape signature" as described above) is *Nt5e* which encodes CD73 (Fig. 7c). CD73 is a marker of naturally occurring CD4 anergic cells together with FR4[78]. As with HELIOS, CD73 is induced in a large subpopulation (>20%) of DKO OT-II SP4 (Foxp3-neg) only in RIPmOVA but not WT hosts (Fig. 7h, Supp

Fig 9a, b). By contrast, the analogous WT OT-II cell population in RIPmOVA hosts is largely absent, consistent with their deletion.

Similar to RIPmOVA model, polyclonal Foxp3-neg DKO SP4 also harbor a large (>20%) CD73hi population that are also HELIOS+ (Fig. 7i-k). This compartment far exceeds the size of a WT tTreg population and is apparent despite a diverse repertoire of T cell clones and endogenous antigens. We postulate that the appearance of this population reflects escape of DKO cells from negative selection by

**Fig. 5 | Transcriptional program induced in thymocytes fated for deletion is conserved in normal thymic repertoire and enriched for anergy-associated gene. a** Post-sort semimature CD25- SP4 thymocytes from Nur77-eGFP reporter gated as in Supp Fig. 8a and sorted according to lo, med, hi GFP expression, overlaid on pre-sort population. Biological triplicate samples subjected to bulk RNAseq analysis (GSE279344 and Supplementary Data 3a). **b** PCA plot of RNAseq analysis for samples in (**a**). **c** TPM values from GFPlo, med, hi SP4 samples sequenced as in (**a**) are plotted for GFP, Nr4a1-3, and Bcl2l11. These data represent biological triplicate samples subjected to bulk RNAseq analysis. Statistical analysis in Supplementary Data 3a. **d** Heatmap depicts normalized expression (z-score) of all genes with significant differential expression ( | log2FC | ≥ 1, adj p value/FDR ≤ 0.05) in any pairwise comparison. Genes are ordered by log2FC. **e** Volcano plot depicts significantly differentially expressed genes ( | log2FC | ≥ 1, FDR ≤ 0.05) between Nur77-GFPhi and -GFPlo CD4 T cells in black, while genes not passing significance thresholds are shown in light gray. Genes of interest are highlighted in green and annotated with their gene symbol. (GSE279344 and Supplementary Data 3a show Deseq2 analysis). **f** Gene sets from two differential expression analyses – OT-II from RIPmOVA hosts vs. WT hosts, and CD4 T cell Nur77-GFPhi vs. GFPlo – were each filtered for FDR ≤ 0.05 and for genes appearing in both sets. FC

FC plot depicts genes with |log2FC| ≥ 1 in at least one gene set in black, while genes with |log2FC| ≤ 1 in both gene sets are shown in light gray. Genes of interest are highlighted in green and annotated with their gene symbol. See also Supplementary Data 3b. **g** Gene set enrichment analysis (GSEA) was undertaken to compare the list of genes upregulated in OT-II thymocytes from RIPmOVA hosts relative to WT hosts (log2Fc ≥ 1, fdr <0.05) to the genome-wide data set of TPM in Nur77-eGFP-hi and lo semimature SP4 as in (**a**) above (GSE279344). **h** Heatmap depicts dataset GSE178782 encompassing transcriptome of naïve (CD62Lhi CD44lo FR4lo CD73lo CD25-) CD4 T cells of WT polyclonal genotype +/- 3 hr anti-CD3 stimulation as well as ex vivo naturally occurring anergic CD4 T cells (CD62Llo CD44hi FR4hi CD73hi CD25-). Genes included in the heatmap are those identified in our data set (Fig. 5a) that are DEG in OT-II thymocytes in RIPmOVA hosts relative to WT hosts (log2FC ≥ 1, fdr<0.05, N = 286). Modules depict selected genes that are either primary response genes, anergy-associated genes, or naïve markers. See also Supplementary Data 4a. **i, j** GSEA was undertaken to compare either (**i**) genes upregulated in OT-II from RIPmOVA hosts (as in g above), or (**j**) genes upregulated in Nur77-eGFP-hi vs. -lo SP4 (log2FC ≥ 1, padj> 0.05 against genome-wide data set of TPM in naïve and anergic WT CD4 T cells as in (**h**) above, (GSE178782).

endogenous antigen recognition. Strikingly, among polyclonal DKO SP8 thymocytes, CD73hi population was also uniquely expanded and constituted nearly half of all SP8s (Fig. 7l-n). This suggested that a larger fraction of SP8 than SP4 are rescued from deletion, consistent with a more stringent threshold for deletion in that lineage[48,49]. This moreover suggests that altered SP4/SP8 ratio in DKO reflects differential threshold for deletion rather than altered lineage specification. Importantly, both OT-II and polyclonal DKO CD73hi cells express CD24 and CD62L demonstrating that they are indeed bona fide thymocytes rather than recirculating mature T cells[45,77].

### DKO thymocytes exhibit an anergic imprint that persists in the periphery

WT OT-II CD4 splenic T cells are almost entirely lost in RIPmOVA hosts (Fig. 2j Supp Fig 9c). Strikingly, not only do DKO OT-II cells escape deletion in the thymus and populate the periphery, but they harbor a large CD73hi sub-population that is evident in the spleen early after reconstitution (Fig. 8a-c) This corresponds to a similar population concurrently detected among DKO OT-II SP4 thymocytes (Fig. 7e-h, Supp Fig 9a, b), suggesting they may represent the same cells captured after thymic egress. Because this phenotype is evident in splenic naïve T cells while RIPmOVA is tissue-restricted, we hypothesize that this reflects thymic rather than peripheral self-antigen encounter.

DKO T cells from polyclonal chimeras also accumulate relative to WT in the periphery of competitive chimeras and express anergy-associated phenotypic markers at an early time point in similar proportion to DKO thymocytes (Fig. 8d-f, Fig. 7i-n, Supp Fig 9d-i). These polyclonal T cells are CD73hi, co-express FR4, and a subset are also HELIOS+ (Fig. 8e, f, Supp Fig 9f, h). Importantly, DKO CD73hi HELIOS+ T cells from both polyclonal and OT-II chimeras are Foxp3-negative and exhibit a naïve CD62Lhi CD44lo surface phenotype (Supp Fig 9c-e), similar to PopD (see Fig. 6)[74,75] but distinct from a previously described CD44hi CD73hi FR4hi anergic population [78]. Moreover, these CD73hi naïve CD4 T cells are also functionally anergic – as evidenced by impaired induction of pErk following in vitro TCR stimulation (Fig. 8g, h). We propose that this cell-intrinsic, anergy-like state among peripheral naïve T cells is induced in the thymus, marked by CD73 and HELIOS expression, and represents an alternate or fail-safe fate for highly self-reactive DKO thymocytes that evade deletion and diversion. Interestingly, the most self-reactive WT T cells with high CD73 expression have a modest version of the same phenotype, Fig. 8g, h; p = 0.006).

### Nr4a required to maintain tolerance to TSA

RIPmOVA – but not WT – host chimeras harboring both OT-II and DKO OT-II donors (schematic Fig. 2a) develop elevated blood sugar

consistent with diabetes (Supp Fig 9j). Indeed, representative pancreatic sections reveal islet immune infiltration and, in some cases, frank destruction (Fig. 8i, j). Because WT OT-II T cells are efficiently deleted in RIPmOVA hosts while DKO OT-II T cells escape to the periphery (Fig. 2j), we hypothesize that DKO OT-II peripheral T cells promote disease in these chimeras. Importantly, the majority of peripheral Treg in these chimeras – normally reconstituted – are of polyclonal donor origin as seen during thymic development (Fig. 2e, f, Supp Fig 9c), while DKO OTII T cells in the periphery of RIPmOVA hosts are Foxp3-. To isolate contribution of WT OT-II and DKO OT-II cells in disease pathogenesis, we sought to track these genotypes in separate RIPmOVA hosts. To this end, we generated a distinct set of chimeras in which either WT OT-II or DKO OT-II donors were mixed with excess of polyclonal BM. This mixture was transferred into two distinct set of irradiated RIPmOVA hosts (see schematic, Supp Figure 10a). Similar to competitive OT-II chimeras described earlier (Fig. 2), WT OT-II cells in RIPmOVA recipients were counter-selected relative to polyclonal T cells and the few remaining cells were largely diverted into the Treg compartment, while DKO OT-II were neither deleted nor diverted, and evaded TCR downregulation as well (Supp Fig 10b–k). We identified islet inflammation in RIPmOVA recipients of DKO OT-II but not WT OT-II cells (consistent with lack of disease in prior reports[58]), suggesting Nr4a1/3 are essential to maintain immune tolerance to the model TRA RIPmOVA (Fig. 8k, l, Supp Fig 10h).

## Discussion

Deletion of self-reactive thymocytes is a vital central T cell tolerance mechanism, yet the molecular pathways that impose this fate have been elusive. Here we identify Nr4a1 and Nr4a3 as essential factors that link self-antigen recognition to a transcriptional program that imposes not only clonal deletion and Treg diversion but also a cell-intrinsic anergy-like tolerance program in developing thymocytes.

Although *Nr4a1* and *Nr4a3* exhibit functional redundancy, their regulation and expression diverge in critical ways. Prior studies suggest that the half-life of Nr4a1/Nur77 protein is shorter than that of Nr4a3/Nor1[28,79](immpres.co.uk; Supp Fig. 1b). This may help explain functional redundancy with *Nr4a1* despite low *Nr4a3* transcript abundance. The rapid turnover of *Nr4a1*/Nur77 protein[28,31,32] may prevent substantial accumulation of *Nr4a1*/Nur77 protein following the brief, intermittent TCR signals during positive selection. However, *Nr4a1* accumulation is more likely following sustained TCR-APC contacts (> 3 hrs) shown to be required for deletion[35,80]. Induction of *Nr4a2* and *Nr4a3* is exclusively dependent on NFAT and exhibits a higher TCR signaling threshold compared to *Nr4a1*[81]. Expression of *Nr4a3* in addition to *Nr4a1* may tip the balance towards deletion in response to strong TCR signaling. We postulate that the Nr4a family may function

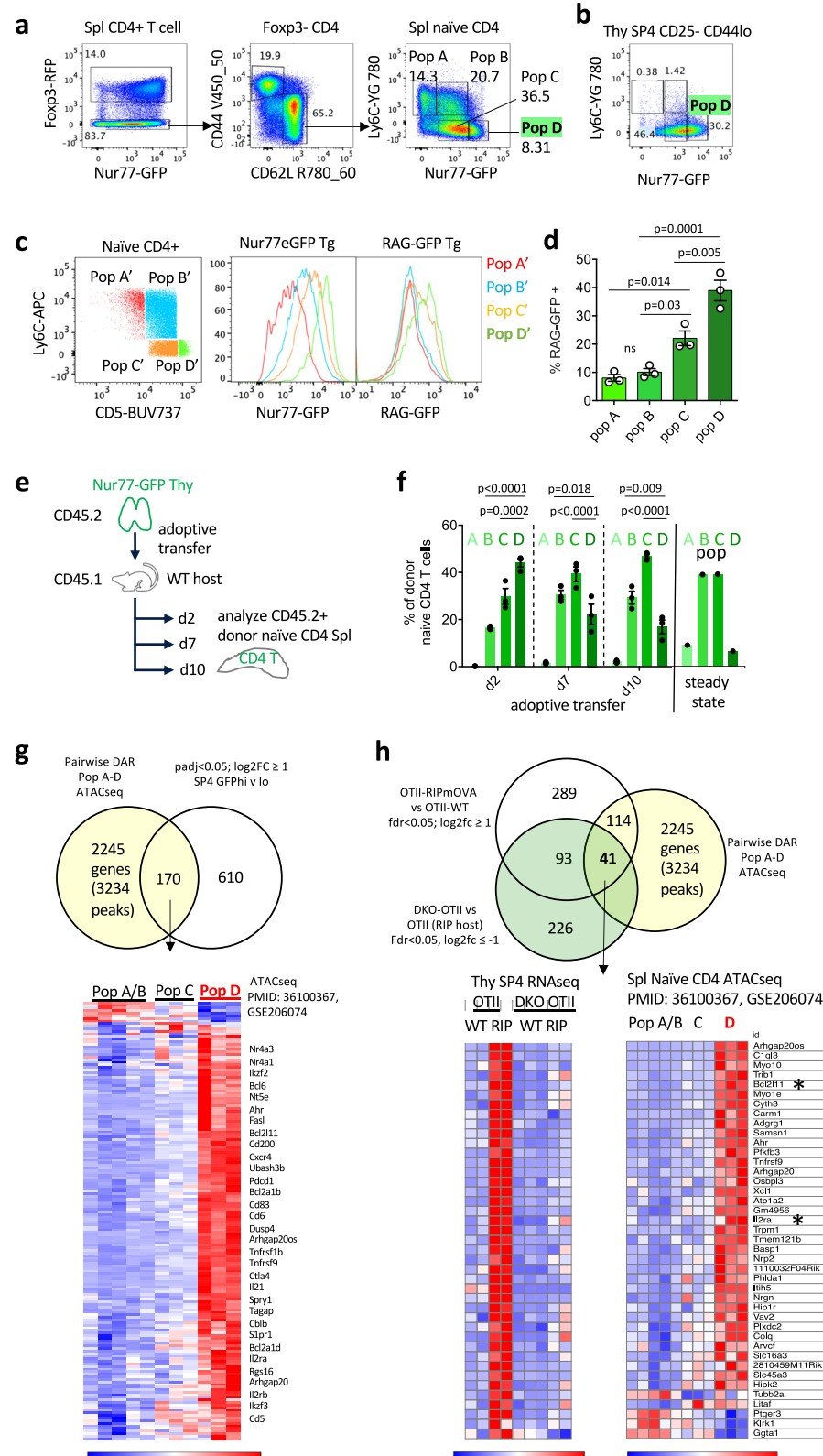

as a high pass filter to translate only sustained, high affinity pMHC interactions into tolerogenic fates.

The Nr4a family has long been implicated in Ag-induced death of lymphocytes, but the mechanism remains controversial. In a proposed post-translational mechanism, Nur77/Nr4a1 induces a conformational change in Bcl2 that exposes its BH3-only domain, thereby transforming this anti-apoptotic factor to a "killer"[26]. This shifts the balance between pro- and anti-apoptotic Bcl2 family members, triggering caspase-induced apoptosis. However, evidence for this pathway during thymic negative selection is circumstantial and limited to DP thymocytes[27,82]. A transcriptional mechanism was alternatively proposed because truncated Nr4a1 constructs that retain their DNA-binding domain rescue deletion and suppress transcription by other Nr4a family members in a dominant negative

**Fig. 6 | Epigenetic imprint of thymic antigen encounter persists in recent thymic emigrants with "anergic" phenotype. a, b** Gating scheme to identify Pop A - D among naïve CD4 splenic T cells (**a**) from mice harboring both Foxp3-RFP reporter and Nur77-eGFP reporter or CD25- SP4 thymocytes from Nur77-eGFP reporter on basis of Ly6C and GFP (**b**). Sequential gating in (**a**) excludes Foxp3+ Treg and defines Pop A-D among naïve CD62Lhi CD44lo T cells. **c** Gating scheme to identify analogous Pop A'-D' on basis of Ly6C and proxy marker CD5. Histograms depict GFP expression in PopA'-D' as gated in lefthand plot in either Nur77-GFP reporter (middle histograms) or RAG-GFP reporter (righthand histograms). **d** quantification of RAG-GFP+ fraction +/- SEM of Pop A'-D' (as gated in **c**) from N = 3 biological replicate RAG-GFP animals. **e, f** schematic (**e**) depicts experimental design with adoptive transfer of thymocytes from Nur77-eGFP reporter mice into congenically marked CD45.1+ hosts. At sequential time points after transfer, recipient splenocytes were harvested. Following enrichment for CD4+ cells by benchtop negative selection, splenocytes were stained to detect cells of donor origin among CD25- naïve CD4 T cell in Pop A-D (gating as in Supp Fig. 8b). Graph (**f**) depicts quantification of distribution across pop A-D of donor cells at sequential time points for N = 3 biological replicates +/- SEM. **g** Venn diagram depicts overlap between genes differentially expressed in Nur77/Nr4a1-hi vs. Lo SP4 semimature thymocytes (see Fig. 5a, GSE279344) and genes near differentially accessible regions (DAR) in the genome across Pops A-D (PMID: 36100367; GSE206074). To do so, DAR were collapsed by nearest gene. Heatmap depicts rpm in Pops A-D for most differentially accessible OCR near these genes (see also Supplementary Data 4e). **h** Venn diagram as in (**e**) but further filtered to identify Nr4a-dependent gene set. Heatmap depicts expression of those genes in our thymic RNAseq data set aligned against accessibility in Pop A-D ATACseq (PMID: 36100367; GSE206074; see also Supplementary Data 4f). Statistical analysis: **d** One-way Anova with Tukey's multiple comparison correction. data in **a**–**d** are representative of at least N = 4 biological replicates. **f** Two-way Anova with Holm-Sidak multiple comparison correction. Source data are provided as a Source Data file.

manner[11,15,83]. Here we show that the Nr4a family function definitively in a transcriptional ("genomic") capacity to regulate *Bcl2l11*/BIM in self-reactive SP thymocytes[62].

*Bcl2l11*/BIM and *Nr4a1*/Nur77 are coordinately upregulated in strongly signaled post-selection DP thymocytes in proportion to self pMHC reactivity. BIM has been implicated in an early wave of deletion at this stage, although this role varies according to model system[10,63,67]. Dominant negative *Nr4a1*/Nur77 Tg constructs were reported to interfere with deletion driven by superantigen and ubiquitous model Ag[14,83]. We observe only a subtle selective advantage for polyclonal DKO thymocytes in the post-selection DP compartment, in contrast to *Bcl2l11*/BIM-/- phenotype[64], suggesting a selective role for Nr4a1/3 in late deletion.

Expression of *Nr4a* genes and *Bcl2l11*/BIM are also induced during a second wave of negative selection triggered by self-Ag recognition in medullary SP thymocytes. BIM (along with PUMA) is essential for TRA-induced deletion in the medulla[10,42,63,67]. We identify Nr4a consensus motifs and an Nr4a1/Nur77 ChIPseq peak in an enhancer required for *Bcl2l11* induction in the thymus whose deletion phenocopies *Bcl2l11*/BIM-/- and rescues negative selection[68]. We also identify a profound competitive advantage for *Nr4a1/3* DKO OT-II and polyclonal thymocytes at the SP stage that coincides with a complete failure of DKO thymocytes to upregulate *Bcl2l11*/BIM in response to endogenous self-Ag and in vitro TCR stimulation. In contrast, *Nr4a1* SKO OT-II thymocytes express subtly and non-significantly reduced levels of *Bcl2l11* transcript[17], while *Nr4a1* SKO OT-I thymocytes show no difference in either BIM protein expression or deletion[20]. In addition, we show here that *Nr4a3* SKO OT-II thymocytes exhibit no defect in robust BIM induction in RIPmOVA hosts. Together, these results reveal for the first time nearly complete functional redundancy between *Nr4a1* and *Nr4a3* in BIM induction and negative selection of SP thymocytes that is uniquely unmasked by deletion of both family members. Taken together, our data are consistent with a model where strong TCR signaling in the medulla induces coordinate upregulation of Nr4a1 and Nr4a3, which trigger negative selection through transcriptional upregulation of *Bcl2l11*/BIM.

An extensive literature demonstrates that Treg fate is agonist-selected by TRAs expressed in the medulla and can emerge at the boundary of the TCR signaling threshold for clonal deletion[5]. A recently published single cell thymocyte atlas (Steier et al.) reveals that Treg and medullary SP thymocytes fated for negative selection occupy adjacent clusters with similar transcriptomes[41]. For example, *Nr4a1, Nr4a3*, and *Ikzf2*/HELIOS are expressed in both populations, suggesting they mark a common progenitor population of strongly "signaled" SP thymocytes. The fate decision between diversion to Treg and deletion is further influenced by additional signals, such as co-stimulation and IL-2[5]. Indeed, IL-2 supply has long been appreciated as a key and limiting requirement for Treg survival and fate in the thymus, and Steier et al. reveal an IL2/STAT5 module marking this precursor population across a gradient. In addition to the cell-intrinsic requirement for the Nr4a family to promote transition between CD25hi pre-Treg to CD25hi Foxp3+ Treg (evidenced by a partial block at this transition for DKO SP4 thymocytes)[50], we also observed a cell-extrinsic impact of Nr4a-deficiency on Treg fate. Rudensky and colleagues as well as others have suggested strongly-signaled thymocytes are the main source of IL-2 to promote Treg fate[52,54]; we postulate that DKO self-reactive clones that have escaped deletion are a source of excess IL-2 that promotes Treg fate among donor thymocytes of WT origin in the same chimeric thymus. In addition, Nr4a-deficient T cells have been shown by us and others to over-produce IL-2[21,70].

Unexpectedly, in polyclonal DKO chimeras we describe emergence of a unique Foxp3+ SP8 population (as well as SP8 CD25hi Foxp3-neg putative precursors) that are absent in WT chimeras. Foxp3+ SP8 is both a cell-intrinsic and cell-extrinsic phenotype in DKO chimeras, raising the possibility that both escape of DKO self-reactive SP8 from deletion and excessive IL-2 production may contribute to this fate. This may relate to mechanisms driving a Foxp3+ CD8+ peripheral population identified among tumor-infiltrating T cells, as well as in alloreactive T cells from GVHD models where it is constrained by high BIM expression[55,56]. Indeed, we show that semimature SP8 thymocytes express higher levels of both Nur77-GFP and BIM, and are more stringently censored by deletion than SP4 thymocytes[48,49]. Strikingly, in addition to cell-intrinsic Nr4a-dependent BIM regulation, BIM expression is also suppressed in a cell-extrinsic manner among WT SP8 thymocytes in DKO but not control chimeras. We hypothesize that excess IL-2 produced by DKO thymocytes may suppress *Bcl2l11*/BIM in WT cells; indeed IL-2 has been reported to prevent deletion of Foxp3+ T cells, and their survival in *Il2-/-* animals is abrogated by deletion of *Bcl2l11*[53,84]. Taken together, this leads us to propose that Nr4a and IL-2 signals converge to censor a latent Foxp3+ SP8 fate in the thymus.

Importantly, in RIPmOVA hosts, WT OT-II thymocytes are overwhelmingly fated for deletion rather than Treg diversion. Our transcriptomic analyses indicate that neither *Il2ra*/CD25 nor *Foxp3* are substantially upregulated in OT-II cells, and therefore these strongly signaled SP cells likely do not comprise a population of committed pre-Treg - in contrast to prior studies[50]. Rather, we postulate that HELIOS and CD73 expression mark strongly signaled DKO thymocytes that have escaped deletion. A similar HELIOS+ population was identified in the *Bcl2l11*/BIM-/- InsHEL model even after Treg are gated out[42]. Strikingly, we observe a similar proportion of CD73+ HELIOS+ cells among peripheral naïve DKO CD4 T cells at the early time-points examined in this study. These cells exhibit not only phenotypic but also functional features of anergy. We postulate that this anergic 'naïve' peripheral T cell phenotype in DKO cells is induced by thymic rather than peripheral self-antigen encounter and persists for a time after thymic egress.

We discovered that the Nr4a family is an essential molecular link between TCR signaling and a broad transcriptional program induced in

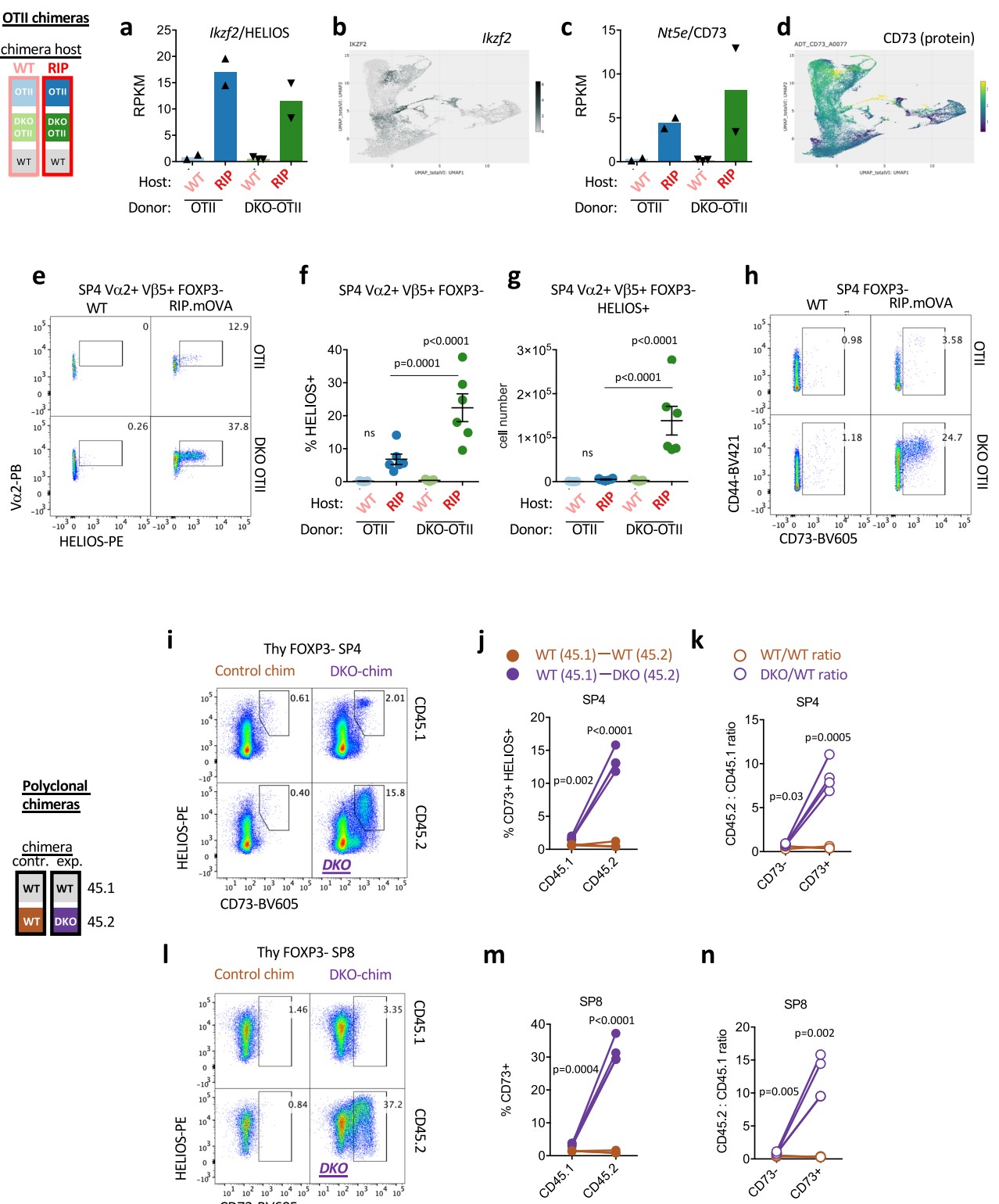

response to TRA. Interestingly, a thymocyte transcriptional defect reminiscent of Nr4a-deficiency has been described in autoimmune-prone NOD mice. NOD thymocytes exhibit impaired *Bcl2l11* induction and escape from deletion triggered by InsHEL model antigen[40,66]. Of the 82 InsHEL-induced genes that overlap with RIPmOVA-induced genes in our data set, over half (56) exhibit a defect in the NOD background. Since this defect in NOD – and DKO – thymocytes is broad but not global, it cannot reflect a proximal defect in TCR signal transduction. Among NOD-dependent genes, we observed a

remarkable enrichment in Nr4a-dependent genes (23/56), including *Eno3, Tnfrsf9, Lad1*, and of course *Bcl2l11* itself. We propose that polygenic defects affecting the Nr4a-dependent thymic transcriptome may confer risk for autoimmunity.

Strikingly, we provide evidence that the same transcriptional anergy-like program induced in DKO OTII thymocytes that escape clonal deletion is also induced among self-reactive SP4 thymocytes in the natural repertoire (marked by high expression of Nur77/Nr4a1-eGFP reporter). A small number of genes in this shared program correspond

**Fig. 7 | Nr4a-deficient thymocytes escaping deletion acquire an Ag-dependent anergic program in the thymus. a–d** Panels depict *Ikzf*(a, b) and *Nt5e*/CD73 (c, d) expression in our RNAseq data (a, c) and VISION CITEseq (b, d) data as described earlier. RPKM graphs a, c show transcript abundance in CD69hi Va2 + SP4 thymocytes of either WT OT-II or DKO OT-II donor genotypes from chimeras depicted in 3a. (Supplementary Data 1a depicts p value and fdr for all pairwise comparisons across sample types, GSE235101). UMAP data (b, d) from publicly available VISION thymus CITEseq interface. **e–g** Representative plots (**e**) depict gating to identify HELIOS+ cells among SP4 Foxp3- OT-II thymocytes of each genotype from RIP-mOVA and WT host chimeras (see 3a schematic). Graphs depict % (**f**) and absolute number (**g**) of HELIOS+ cells as gated in (**e**) +/- SEM. **h.** Representative plots depict gating to identify CD73+ cells among SP4 Foxp3- OT-II thymocytes of each genotype from RIPmOVA and WT host chimeras (see 3a schematic). Quantification is shown in Supp Fig 9a, b. **i–n** Representative plots depict gating among SP4 (**i**), and SP8 (**l**) subsets to identify HELIOS + CD73+ cells of each donor genotype from experimental DKO:WT polyclonal chimeras and from control WT:WT polyclonal chimeras (see 1c schematic). **j, m** Quantification of % CD73/HELIOS gates from each donor genotype as shown in (**i, l**). Lines connect donors in the same chimera. **k, n** Quantification of ratio of donor genotypes CD45.2/CD45.1 from each chimera among CD73/HELIOS in panels **i, l** Statistical tests: **f** one way ANOVA with pre-specified comparisons corrected for by Sidak test. **g** one way ANOVA with Tukey's multiple hypothesis test. **j, k, m, n,** unpaired, parametric, two-tailed t-test, assume equal SD. Graphs in **f, g**, depict N = 7 WT host and N = 6 RIPmOVA host chimeras – biological replicates. Graphs in **j, k, m, n** depict N = 3 WT:WT control chimeras and N = 4 DKO:WT chimeras – biological replicates and are representative of 3 independent sets of chimeras. Source data are provided as a Source Data file.

to primary response genes induced by TCR stimulation in peripheral T cells, but the majority of transcripts – including many negative regulators of T cell signaling - require chronic antigen stimulation for their expression in the periphery. We propose that this transcriptional program may contribute to an anergy-like state in the thymus. This "anergy" signature includes genes upregulated in exhausted CD8 T cells and tolerant CD4 T cells (signal 1 without signal 2), as well as NFAT-induced genes. Notably, the Nr4a family contributes to the anergy/exhaustion transcriptome and epigenome of peripheral T cells[21,70,85], and here we show that they may contribute to a related transcriptional program of non-deletional tolerance in the thymus as well.

Functional unresponsiveness as an alternative fate to deletion in the thymus has been previously proposed; classic papers showed this for class I-restricted TCRs[86] or class II-restricted TCRs[87] that escape deletion by superantigen (Mls-1a), and also for TEC expression of an alloantigen driven by keratin IV promoter[88]. More recently a tetramer approach was used to demonstrate a variety of fates for self-reactive T cells (deletion, diversion, and 'ignorance') that depended on location and amount of self-antigen presentation in the thymus[89]. Elimination of negative selection has also unmasked alternate fates for self-reactive lymphocytes[90]. *Bcl2l11*/BIM-deficient T cells, which escape deletion by TECs yet do not produce autoimmune disease[63,67], express anergic phenotypic markers CD73/FR4 in the periphery[64], and exhibit impaired TCR signaling[91]. However, it has not been clear in most prior studies if clonal anergy of cells evading deletion was imposed in the thymus or only in the periphery where it has long been appreciated to play a vital role. Most recently, Baldwin and colleagues showed that non-deletional control of OTI *Bcl2l11*/BIM-/- thymocytes from RIPmOVA hosts is associated with PD-1 upregulation in the thymus[92]. We propose that *Pdcd1*/PD1 induction in response to self-antigen is part of a larger transcriptional program we identify in RIPmOVA OT-II thymocytes that is partly mediated by the Nr4a family. Our DKO chimera model uniquely disables both Treg diversion and deletion to unmask this tolerance program in the thymus and reveals the contribution of the Nr4a1/3 to its establishment.

Naturally occurring naïve (CD62Lhi) peripheral T cells marked by high Nur77-eGFP expression exhibit impaired TCR signaling and IL-2 production, and an increased propensity to differentiate into Treg[74,75]. Strikingly, we provide evidence to show that this population is enriched for recent thymic emigrants and also harbors an epigenetic imprint that corresponds to the Ag-induced and Nr4a-dependent transcriptional program we identified in self-reactive SP thymocytes both in the OTII model and in the normal repertoire. Expression of inhibitory co-receptors such as *Pdcd1*/PD-1, *Ctla4*, *Lag3*, and *Cd200* in this thymically-imprinted RTE population might render these cells particularly vulnerable to checkpoint blockade. Consistent with this idea, it was recently shown that thymoma and thymic remnant size constituted major risk factors for development of checkpoint myocarditis and myositis[93]. The anergy-like transcriptional program we describe in self-reactive thymocytes may represent a poised state that

operates to preserve tolerance at the boundary of Treg diversion and deletion in the thymus and can persist among RTEs after thymic egress– enabling clones that encounter peripheral self-antigen to undergo diversion, deletion, and anergy[94]. Anergic CD73hiFR4hi T cells in the CD44hiCD62Llo memory compartment are also poised to differentiate into Treg, suggesting a transcriptional and epigenetic link between anergy and Treg persists throughout T cell ontogeny[78].

By transmitting a broad TCR-induced transcriptional program in response to TRA recognition in the medulla, the Nr4a family not only mediates Treg diversion as previously shown, but also drives BIM-dependent deletion, and contributes to a non-deletional anergy-like tolerance program in the thymus that can persist in the normal repertoire of peripheral T cells.

## Methods

### Mice

All mice were housed in a specific pathogen-free facility at UCSF or Emory University according to institutional and National Institutes of Health guidelines. C57BL/6 (CD45.2) and BoyJ (CD45.1) mice (both considered WT here) were initially purchased from The Jackson Laboratory (Strain #:000664) or Charles River Labs (Strain Code 564), respectively. Nur77-eGFP BAC Tg mice were previously described[29]. RAG-GFP mice were previously described and characterized as marker of RTEs[77,95]. OT-II TCR transgenic mice[96] and RIP.mOVA transgenic mice[57] were previously described. *Nr4a3*-deficient allele was generated in our laboratory on the C57BL/6 genetic background as previously described[79]. *Nr4a3-/-Nr4a1-/-* (DKO) mice were previously described[21] and were bred to OT-II mice to create the OT-II *Nr4a1-/-Nr4a3-/-* strain. BIM-deficient B6.129S1-*Bcl2l11*^tml.1Ast/J line (Jackson Strain #:004525)[10] were crossed to Nur77-GFP mice. Foxp3-RFP[97] mice crossed to Nur77-GFP were previously described[75]. All strains were fully backcrossed to C57BL/6 genetic background for at least 6 generations. RIP.mOVA CD45.1/2 and WT CD45.1/2 mice were F1 progeny from RIP.mOVA x BoyJ crosses. OTII was crossed with BoyJ to generate and maintain OTII CD45.1 line. Mice of both sexes were used for experiments between the ages of 3 and 10 weeks except for BM chimeras as described below. Experimental and control chimeras were co-housed following irradiation. All experiments were approved by UCSF and Emory Institutional Animal Care & Use Committees (IACUC). Standard housing conditions used time controlled lighting on standard 12:12 light:dark cycle; humidity between 30-70%; temperature 68-79 degrees F. Euthanasia was carried out using approved protocols: carbon dioxide inhalation followed by cervical dislocation.

### Antibodies and reagents

Abs for surface markers: Abs to CD4 (RM4-5 or GK1.5), CD5 (53-7.3), CD8 (53-6.7), CD24 (M1/69), CD25 (PC61.5), CD62L (MEL-14), CD69 (H1.2F3), CD44 (IM7), CD45.1 (A20), CD45.2 (104), CD73 (TY/11.8), FR4(12A5), Ly-6C(AL-21), Vα2 (B20.1), Vβ5(MR9-4), and TCRβ (H57-597) conjugated to fluorophores were used (BioLegend,

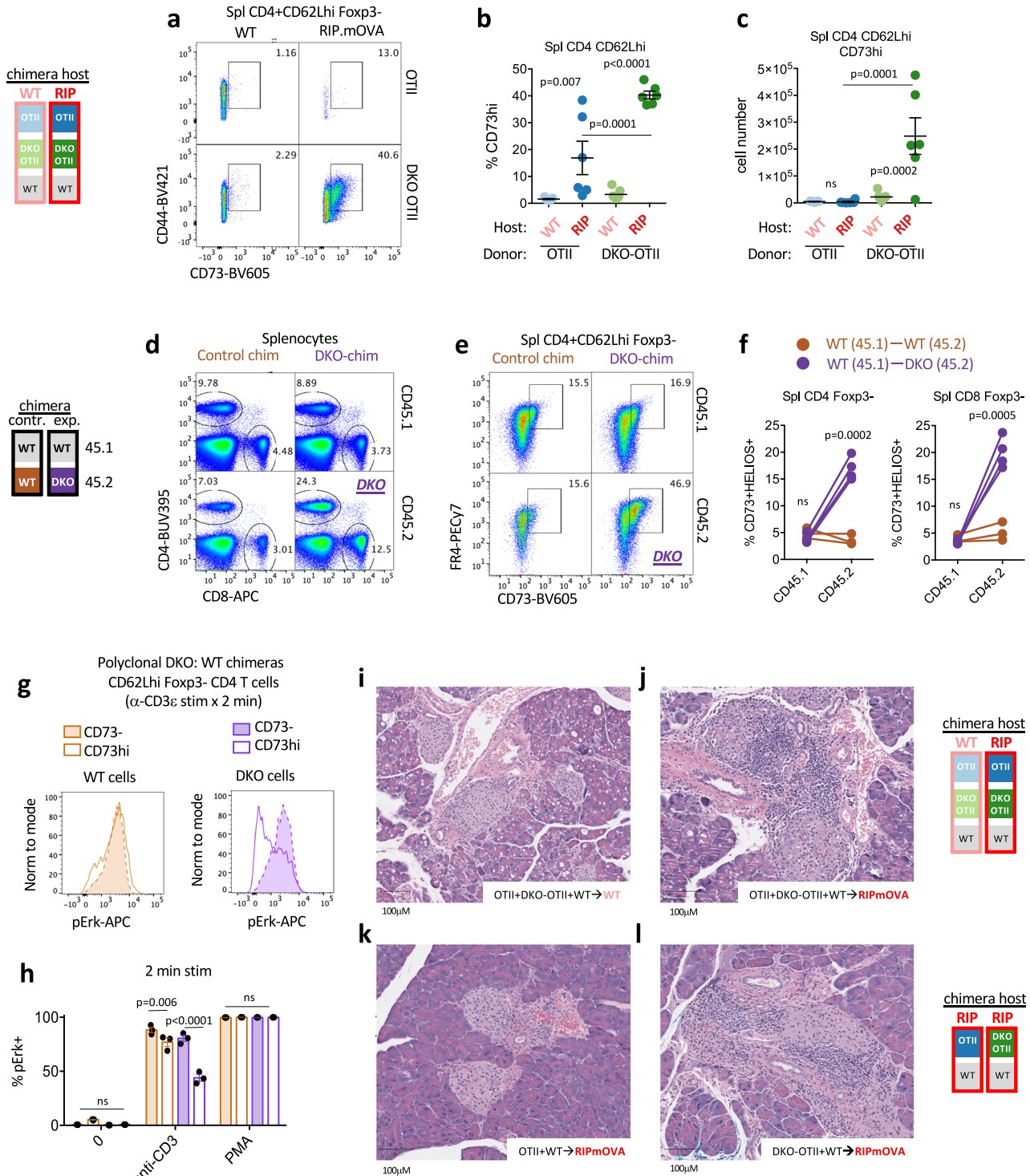

eBiosciences, BD, or Tonbo). Abs for intra-cellular staining: HELIOS Ab conjugated to PE (clone 22F6, cat. 563801, BD Biosciences), BIM (clone C34C5; cat. 2933S, Cell Signaling) Rabbit mAb, FOXP3 Ab conjugated to APC or FITC (clone FJK-16s, Invitrogen). Anti-pERK (Phospho-p44/42 MAPK (T202/Y204) (clone 197G2, Cell Signaling) Rabbit Ab. Donkey Anti-Rabbit IgG (H + L) conjugated to APC (Jackson ImmunoResearch).

Stimulatory Abs: Anti-CD3ε (clone 2c11; BioLegend cat: 102102). Anti-CD28 (clone 37.51, BioLegend cat: 102102). Goat anti-Armenian Hamster antibody (Jackson ImmunoResearch).

Culture Media: RPMI-1640 + L-glutamine (Corning-Gibco), Penicillin Streptomycin L-glutamine (Life Technologies), HEPES buffer [10 mM] (Life Technologies), B-Mercaptoethanol [55 mM] (Gibco), Sodium Pyruvate [1 mM] (Life Technologies), Non-essential Amino acids (Life Technologies), 10% heat inactivated FBS (Omega Scientific).

**Flow cytometry**
Cells were analyzed on a Fortessa and sorted on Aria (BD Biosciences). Data analysis was performed using FlowJo (v10) software (BD Biosciences).

**Fig. 8 | DKO T cells acquire an anergic imprint in the thymus that persists in the periphery, but break tolerance. a–c** Representative plots depict gating to identify CD73+ cells among Foxp3- CD4 naïve T cells of each donor genotype (OT-II or DKO OT-II) from RIPmOVA and WT host chimeras (see 2a schematic). Graphs depict % (**b**) and absolute number (**c**) of CD73+ cells as gated in (**a**) +/- SEM. **d** Representative plots depict gating to identify splenic CD4 and CD8 T cells of each donor genotype from polyclonal chimeras (see Fig. 1c schematic). See supp Fig 9g for quantification. **e** Representative plots depict gating to identify CD73+ cells among Foxp3- CD4 naïve T cells of each donor genotype from polyclonal chimeras (see Fig. 1c schematic). **f** Graphs depict % CD73 + /HELIOS+ cells as gated in Supp fig 9f. from each donor genotype in polyclonal chimeras (see Fig. 1c schematic). Lines depict donor genotypes from the same chimera. **g, h** Splenocytes from polyclonal DKO:WT chimeras (as in schematic Fig. 1c except 14 week reconstitution) were stimulated with anti-CD3/cross-linking Ab for 2 minutes, followed by fixation, permeabilization and staining to identify intracellular pErk as well as congenic markers CD45.2/CD45.1, Foxp3, CD44, CD62Lhi, CD4, and CD73/FR4 to identify naïve Foxp3- CD4 T cells with either high or low CD73 expression of each

donor genotype. Representative overlaid histograms depict pErk in these population from either CD45.1 WT (brown) or CD45.2 DKO (purple) cells. Graph depicts quantification of pErk+ gate across these populations from N = 3 biological replicates +/- SEM. **i, j** representative histology from WT (**i**) or RIPmOVA (**j**) host chimeras containing donor WT OT-II and DKO OT-II and polyclonal WT BM as schematized (see also Fig. 2a chimera design) at 6 weeks following irradiation/BM transfer. **k, l** representative H&E pancreas histology from RIPmOVA host chimeras containing either OT-II:WT-polyclonal or DKO OT-II:WT-polyclonal donor BM as schematized (see Supp Fig 10a chimera design) at 10.5 weeks following irradiation/BM transfer. Statistical tests: **b**, **c** one way ANOVA with Tukey's multiple hypothesis test. Graphs in b, c, depict N = 7 WT host and N = 6 RIPmOVA host chimeras. **f** unpaired parametric two-tailed t-test, assume equal SD. Graphs in **f** depict N = 3 WT:WT control chimeras and N = 4 DKO:WT chimeras. representative of at least 3 independent sets of chimeras. **h** ordinary two-way ANOVA with Tukey's multiple comparisons test. Data in **i**, **j** are representative of 7, 6 biological replicates respectively. Data in **k**, **l** are representative of 3 biological replicates each. Source data are provided as a Source Data file.

## FOXP3/HELIOS staining
FOXP3 and HELIOS staining was performed utilizing a FOXP3/transcription factor buffer set (eBioscience) in conjunction with APC or FITC anti-FOXP3, as per manufacturer's instructions.

## BIM staining
Intracellular staining for BIM was performed with secondary Donkey anti-rabbit APC secondary antibody using BD cytofix/cytoperm kit (BD biosciences 554722) per manufacturer's instructions. For in vitro stimulation of thymocytes, 96 well plates were precoated overnight with 10 µg/ml anti-CD3ε (clone 2c11) and 2µg/ml anti-CD28 (clone 37.51) in PBS. After washing, 1.5×10^6 thymocytes/ 200uL C10 from polyclonal chimeras (WT:WT or DKO:WT) were plated into pre-coated or empty wells for 6 hr incubation. At harvest, cells were stained with congenic markers CD45.1 and CD45.2, CD4, CD8, CD69 and BIM as above to detect thymic subsets.

## Live/dead staining
LIVE/DEAD Fixable Near-IR Dead Cell Stain kit (Invitrogen). Reagent was reconstituted in DMSO as per manufacturer's instructions, diluted 1:1000 in PBS, and cells were stained at a concentration of $1 \times 10^6$ cells /100 µl on ice for 15 minutes.

## Bone marrow chimeras
Chimeras were generated by irradiating hosts with either a Cesium source (2 doses of 530 cGy, 4 hours apart) or X-ray source (2 doses of 450 cGY, 3-4 hours apart) followed by rescue with by adoptive transfer of $5 \times 10^6$ total donor BM cells within 24 hours. For each chimeras type, BM from two or three congenically marked donor strains was mixed at specified ratios as depicted in schematics (Figs. 1c, 2a, Supp Fig. 3i, Supp Fig 10a). For polyclonal chimeras generated with X-ray source (Fig. 1c design), both CD45.2 DKO and WT polyclonal donor marrow was depleted of mature T cells by magnetic column-based negative selection (Miltenyi Biotec MACS) using an anti-mouse CD5 biotin-conjugated antibody (Clone 53-7.3, Miltenyi Biotec, 130-101-960) and Streptavidin MicroBeads (Miltenyi BioTec, 130-048-102) per manufacturer instructions. Chimeras were assessed between 5-11 weeks post-reconstitution as noted in figures except for Fig. 8g, h phosflow assays which were conducted with polyclonal chimeras at 14 weeks.

## Phospho-flow
Splenocytes were rested at 37 °C in serum-free RPMI for 30 minutes. Cells were then stimulated with 10 µg/ml of anti-CD3 (clone 2c11) for 30 seconds followed by 50 µg/ml of anti-Armenian hamster cross-linking antibody for 2 minutes, or PMA for 2 minutes. Stimulated cells were fixed with 2% paraformaldehyde and permeabilized with

methanol at −20 °C overnight. Cells were then stained with surface markers and pErk at 20 °C.

## Adoptive transfer of thymocytes
$10^7$ thymocytes (or $5 \times 10^6$ splenocytes) harvested from Nur77-GFP donors (CD45.2 + ) were adoptively transferred via bilateral retro-orbital injection into allotype-marked CD45.1+ BoyJ hosts on d0. At sequential time points, d2, d7, d10, host spleens were harvested, subjected to benchtop MACS negative selection to enrich for CD4+ cells (mouse CD4 kit, Miltenyi Biotec, per manufacturer's instructions), and surface stained with CD25, CD44, CD62L, CD4, CD45.1, CD45.2, and Ly6C to detect donor naïve CD4 T cell subsets.

## Histology
Organs were fixed in 10% formalin for 24 hours, dehydrated progressively, and stored in ethanol at 4 C until processing for H&E by Histowiz.

## Data visualization, analysis, and statistics
Flow cytometry data was analyzed using FlowJo software FlowJo v10.0 Software (BD Life Sciences). Graphs and statistical analyses were undertaken using Prism v6 and matplotlib v3.8.4. All statistical tests were two-sided. All graphs depict SEM except Supp Fig. 1b as noted in figure legends.

## RNA sequencing.
(a) OTII chimeras: DKO OT-II:OT-II:WT - > WT and DKO OT-II:OT-II:WT - > RIP.mOVA bone marrow chimeras were created as described above (Fig. 2a schematic). DKO OT-II (CD4 + CD8-CD69 + Vα2 + CD45.1- CD45.2 + ) and control OT-II (CD4 + CD8-CD69 + Vα2 + CD45.1 + CD45.2 + ) semimature SP4 thymocytes were isolated on BD FACSAria II and BD FACSAria Fusion cell sorters directly into RLT + 1% beta-mercaptoethanol (BME) buffer (Qiagen).

(b) Nur77/Nr4a1-GFP reporter thymocytes: Thymocytes from Nur77/Nr4a1-GFP reporter mice were stained to identify semimature SP4 (CD4 + CD25-CD69hiCD24hi) and 10% of lo, med, hi GFP distribution (see Fig. 5a, Supp Fig. 8a for gates) was sorted as above into RLT + 1% BME buffer in biologic triplicate.

Libraries for both (a) and (b) samples were generated by the Emory Integrated Genomics Core (EIGC) as follows: RNA was isolated using the Quick-RNA MicroPrep kit (Zymo,11-328 M). 2000 cell equivalent of RNA was used as input to SMART-seq v4 Ultra Low Input cDNA Synthesis kit (Takara, 634888) and 200 pg of cDNA was used to generate sequencing libraries with the NexteraXT kit (Illumina, FC-121-10300). Libraries were quantified by qPCR and bioanalyzer traces,

pooled at equimolar ratios, and sequenced on the NovaSeq6000 with a PE100 configuration using a NovaSeq 6000 SP Reagent Kit. For data set (a), raw fastq files were mapped to the mm10 genome using STAR[98] with the GENCODE vM17 reference transcriptome. PCR duplicate reads were marked with Picard MarkDuplicates and removed from downstream analyses. Reads mapping to exons for all unique ENTREZ genes were summarized using GenomicRanges[99] in R v3.5.2 and normalized to reads per kilobase per million (RPKM). For data set (b), raw fastq files were adapter- and quality-trimmed with Trim Galore v0.6.10 and then mapped to the GRCm38 (mm10) genome using STAR v2.7.11b[98] with the GENCODE vM25 gene annotation. PCR duplicate removal and Gene-level counts generation were handled by STAR with the options --bamRemoveDuplicatesType UniqueIdentical and --quantMode GeneCounts, respectively. Counts were normalized to transcripts per kilobase per million (TPM) Differentially expressed genes between sample groups within each of the two data sets were determined using DESeq2[100], and genes that displayed an absolute log2 fold change ≥1= and Benjamini–Hochberg false discovery rate (FDR)–corrected $p$ value ≤ 0.05 were considered significant. PCA plots were made using the plotPCA R function from gene-level counts normalized with variance stabilizing transformation[100].

Heatmaps were generated using the Morpheus online tool (https://software.broadinstitute.org/morpheus/) with default clustering of rows (Pearson minus one) unless otherwise noted in figure legend. Volcano and FC plots were generated using R Statistical Software, Prism, or matplotlib.

### Gene set enrichment analyses
Gene set enrichment analyses were performed with GSEA v4.3.3[97,98] using significantly upregulated gene set generated in this study (absolute log2 fold change ≥1, FDR ≤ 0.05) from OTII cells in RIP.mOVA host vs. OTII cells in WT host (a, above) or from GFPhi vs. GFPlo SP4 thymocytes (b, above). These gene sets were then compared against data set from anergic vs. naive peripheral T cells (GSE178782) or GFPhi/lo SP4 thymocytes.

### Thymocyte CITE-seq data
Steier et al. recently published a single cell mouse thymocyte gene and surface protein expression atlas based on a CITE-seq experiment that included WT polyclonal, MHCII-/-, MHCI-/-, OTII TCR transgenic, and AND TCR transgenic thymocytes (GSE186078)[41]. Here, we present data accessed and analyzed from that experiment via the publicly available VISION graphical user interface (http://s133.cs.berkeley.edu:9002/Results.html) on 5/7/2024. Default settings were used for differential gene expression analyses between noted clusters (as annotated by the authors and shown in Supplemental Fig. 1e and Supplementary Data 2c, d).

### Reporting summary
Further information on research design is available in the Nature Portfolio Reporting Summary linked to this article.

## Data availability
Raw Fastq and processed data files (either RPKM or TPM) for the original RNA-seq analyses have been deposited in the NCBI Gene Expression Omnibus and are publicly available under accession codes GSE235101 and GSE279344, and are also provided in Supplementary Data-Tables 1a, 3a. All previously published/public data sets analyzed in this manuscript are referenced with PMID and data repository locations where available in Supplementary Data-Tables 1-4. All data are included in the Supplementary Information or available from the authors, as are any unique reagents used in this Article. The raw numbers for charts and graphs are available in the Source Data file whenever possible. Source data are provided with this paper.

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

## Acknowledgements

Funding: The Uehara Memorial Foundation Research Fellowship (RH), NIGMS Molecular and Cellular Immunology Program T32AI00733432 (HVN), NIAMS Academic Rheumatology and Clinical Immunology T32 AR079068 (HVN), RRF SDA P0555554 (HVN), UCSF Center for the Rheumatic Diseases (HVN), NIAID R01 AI148487 (JZ), NIAID R01 AI165706 (JZ, BA), NIAID R01AI114575 (JZ). IRACDA NIH Training Award K12GM081266 (AJR). This study was supported in part by the Emory Integrated Genomics Core (EIGC), which is subsidized by the Emory University School of Medicine and is one of the Emory Integrated Core Facilities. We thank Wan-Lin Lo, Ellen Robey, and Arthur Weiss for helpful discussions and critical reading of the manuscript. We thank Alfonse Roque for help with mouse husbandry.

## Author contributions

H.V.N., L.Y., B.B.A., and J.Z. designed the study. H.V.N., L.Y., J.L.M., R.H., B.B.A., and J.Z. executed and analyzed experiments. H.V.N, A.J.R., M.G., and C.D.S. designed and executed bulk RNAseq and

bioinformatic analyses. D.A. assisted with analysis of pre-existing scRNAseq bioinformatic analyses. I.P., E.R., and M.S.A. provided technical support for experiments and data interpretation. H.V.N., B.B.A., and J.Z. drafted and edited the manuscript. L.Y., J.L.M., D.A., and M.S.A. participated in the critical evaluation and editing of the manuscript.

## Competing interests

JZ has been a scientific consultant for Walking Fish, Nurix, Capstan, Sail, and Aramis but there is no direct conflict of interest. The remaining authors declare no competing interests.
