## [Transparent Peer Review file · Nature Communications]

Transcriptional control of central T cell tolerance by NR4A family nuclear receptors

Corresponding Author: Dr Julie Zikherman

Version 1:

Reviewer comments:

Reviewer #3

(Remarks to the Author)

I thank the authors for the impressive number of experiments they have performed to address the reviewers' comments and I stand corrected on some of the concerns I raised. I consider this manuscript suitable for publication in Nature Communications (especially given the vast amount of data presented and the overall solidness of the work), highlighting a few points where I would propose changes or ask for clarifications.

I remain convinced that the presented data indicate that polyclonal DKO precursors give rise to a similar number of thymic CD4⁺Foxp3⁺CD25⁺ thymocytes (Fig2g, I; Rebuttal Fig B, panel a). However it is also clear that in periphery, there are significantly fewer DKO CD4⁺Foxp3⁺CD25⁺ cells (Suppl Fig2g, h). A potential explanation is that in absence of Nr4as (and probably only in presence of WT thymocytes) an unstable nTreg program is executed or these cells have a very short half-life in the periphery. I recognize that the focus of the paper is not on this issue, but as the presence of these thymic nTregs seems to contrast with observations in previous knock-outs I think this should be briefly discussed (maybe also considering the postulated extrinsic regulation of nTreg fate).

In figure 2c, to illustrate the point that WT OTII and DKO OTII thymocytes outcompete polyclonal thymocytes at the 4SP stage in WT recipients and DKO OTII thymocytes outcompete polyclonal thymocytes at the 4SP stage in RIP-mOVA recipients, it would be clearer to first gate on DP and 4SP thymocytes and then assess congenic marker distribution in each of these populations.

I am confused by the experiments with regard to population D. The peripheral D population is enriched in RTEs, as nicely shown by the RAG-GFP experiments (thank you for this). When unsorted RAG-GFP thymocytes are adoptively transferred into recipient mice (Fig 6) over time the thymic, population D-skewed, pattern of Ly6C vs Nur77-GFP expression "redistributes" to the steady state pattern seen in a non-manipulated peripheral naïve T cell repertoire. How is this interpreted? Do population D cells die off or do they change to the B and C populations, and would they in the latter case lose their anergic phenotype? I understand the steady state gene expression analysis of these cells, but can you clarify/argue what the fate of this population would be? Adoptive transfer of sorted, congenically marked population D thymocytes would help clarify this issue.

With regard to the Helios⁺CD73⁺ T cells found in periphery, there is an important difference in response of these cells to TCR stimulation when they are DKO (reduced response) or WT (not very pronounced differences) (Fig 8g). Why this difference and have these DKO cells indeed a correlate in a wild type animal? I consider that in the light of the proposed control by Nr4as of peripheral anergy it is important to discuss the relevance of these findings.

Altogether, I consider that the paper provides a wealth of quality data, unifying, amongst other things, findings from various other studies into a quite coherent frame work. Even though some of the findings, mostly the anergic phenotype T cells, are not completely explained, they are important to be shared with the field.

Point by point response

Reviewer #3 (Remarks to the Author):

I thank the authors for the impressive number of experiments they have performed to address the reviewers' comments and I stand corrected on some of the concerns I raised. I consider this manuscript suitable for publication in Nature Communications (especially given the vast amount of data presented and the overall solidness of the work), highlighting a few points where I would propose changes or ask for clarifications.

We are pleased that the reviewer has found our data convincing.

I remain convinced that the presented data indicate that polyclonal DKO precursors give rise to a similar number of thymic CD4⁺Foxp3⁺CD25⁺ thymocytes (Fig2g, I; Rebuttal Fig B, panel a). However it is also clear that in periphery, there are significantly fewer DKO CD4⁺Foxp3⁺CD25⁺ cells (Suppl Fig2g, h). A potential explanation is that in absence of Nr4as (and probably only in presence of WT thymocytes) an unstable nTreg program is executed or these cells have a very short half-life in the periphery. I recognize that the focus of the paper is not on this issue, but as the presence of these thymic nTregs seems to contrast with observations in previous knock-outs I think this should be briefly discussed (maybe also considering the postulated extrinsic regulation of nTreg fate).

The reviewer is raising two distinct points (which are not the focus of this manuscript):

1. Role of Nr4as in generation of nTreg:

We emphasize the extensive published literature identifying requirement of Nr4a family for normal Treg development (Yoshimura group, PMID: 23334790, PMID: 30089271). We also identified a major defect in nTreg in germline Nr4a1/3 DKO animals previously (PMID: 34343134). Here we again report a substantial defect for DKO nTreg in thymus, but it is not an ASBOLUTE block in development; it is a cell-intrinsic relative block as revealed by comparing precursor and product frequencies between WT and DKO cells in the same chimera. To clarify this point further: The absolute number of DKO nTreg in thymus of DKO:WT chimeras (Fig 1g-I and prior rebuttal figure) is indeed comparable to WT nTreg from control WT:WT chimeras. However, a key point we emphasize is that the relevant comparison to reveal a cell-intrinsic role for the Nr4as in nTreg generation is between DKO and WT cells from the same DKO:WT chimera which share a common milieu. In this setting there remains a substantial DKO defect, reflected in Fig 1h, i. We postulate that a local environment of elevated IL-2 in DKO:WT chimeras is a likely explanation.

Furthermore, Yoshimura group did show that Treg defect is absolute and more severe in TKO thymocytes lacking all three family members Nr4a1, 2, and 3, while defect in DKO thymocytes lacking Nr4a1/3 is less severe and represents a relative rather than an absolute block in nTreg generation (PMID: 23334790: published Fig 3e – see here for convenience).

[Figure redacted]

We believe this collectively accounts for our data, is consistent with previous work, and this is discussed in final edited manuscript.

2. Role of Nr4as in maintenance of Treg:

Supp Fig 2h shows same analysis for peripheral CD4+ Treg. Our interpretation of the data is that peripheral Treg of WT-origin increase as a fraction of WT CD4 compartments compared to proportion in thymus (reflecting either expansion of nTreg and/or pTreg induction of WT cells). By contrast DKO Treg do not increase in the periphery and may in fact decay. This may indeed reflect unstable program. This has been elegantly shown by Yoshimura group using Foxp3-cre to delete Nr4as after Treg generation, demonstrating a role in maintenance (PMID: 26304965). In addition, the same group identified a role for Nr4as in pTreg induction (PMID: 33665581). Therefore, these additional role for the Nr4as may contribute to divergence between DKO and WT Treg abundance in the periphery. We believe this collectively accounts for our data and this is discussed in final edited manuscript.

In figure 2c, to illustrate the point that WT OTII and DKO OTII thymocytes outcompete polyclonal thymocytes at the 4SP stage in WT recipients and DKO OTII thymocytes outcompete polyclonal thymocytes at the 4SP stage in RIP-mOVA recipients, it would be clearer to first gate on DP and 4SP thymocytes and then assess congenic marker distribution in each of these populations.

Supp Fig 3c gating scheme shows exactly this gating scheme for 4SP thymocytes. We have modified this figure to add DP gate as requested, enabling readers to appreciate this point more clearly.

I am confused by the experiments with regard to population D. The peripheral D population is enriched in RTEs, as nicely shown by the RAG-GFP experiments (thank you for this). When unsorted RAG-GFP thymocytes are adoptively transferred into recipient mice (Fig 6) over time the thymic, population D-skewed, pattern of Ly6C vs Nur77-GFP expression “redistributes” to the steady state pattern seen in a non-manipulated peripheral naïve T cell repertoire. How is this interpreted? Do population D cells die off or do they change to the B and C populations, and would they in the latter case lose their anergic phenotype? I understand the steady state gene expression analysis of these cells, but can you clarify/argue what the fate of this population would be? Adoptive transfer of sorted, congenically marked population D thymocytes would help clarify this issue.

To clarify we transferred Nur77-GFP, not RAG-GFP to recipient mice in Fig 6 (I suspect reviewer appreciates this and this was typo above). Our data indeed does not fully determine extent to which initial Pop D cells die or convert to Pop B/C. There is likely some contribution of cell death as overall survival of transferred SP4 thymocytes declines over our time course (see Supp Fig 8c). Co-corresponding author Byron Au-yeung has previously performed the experiment proposed by the reviewer above and data are depicted in Fig 1g, h of PMID 36100367. In that study, Nur77-eGFP lo or hi SP4 thymocytes were sorted and

adoptively transferred. Phenotype of splenic naïve CD4 T cells was assessed after 2 weeks. This shows that while GFP^{hi} SP4 retain GFP levels higher than GFP^{lo} at this time point, there is also substantial redistribution from Pop D to Pop B/C. In other words, both processes contribute. **We have added discussion of this point to the manuscript.**

With regard to the Helios⁺CD73⁺ T cells found in periphery, there is an important difference in response of these cells to TCR stimulation when they are DKO (reduced response) or WT (not very pronounced differences) (Fig 8g). Why this difference and have these DKO cells indeed a correlate in a wild type animal? I consider that in the light of the proposed control by Nr4as of peripheral anergy it is important to discuss the relevance of these findings.

This is a very interesting point. We interpret this to primarily reflect the different repertoire among CD73⁺Helios⁺ DKO cells in periphery relative to WT cells with this phenotype because DKO cells have evaded deletion and are much more self-reactive. Because transcriptional anergy program scales with intensity of TCR stimulation (see new Fig 5a-d), the DKO cell exhibit reduced signaling relative to WT of same CD73⁺ phenotype. Indeed, Nr4as also play a role in peripheral tolerance (PMID: 30814732, PMID: 30814730). We postulate that while Nr4as are required for part of anergy program, the profound self-reactivity and remainder of anergy program account for reduced signaling observed in DKO cells in periphery. We do not think DKO peripheral T cells correspond directly to comparable WT clones, as the DKO population is composed to a substantial extent by cells destined for deletion in WT. However, it may be possible that the most self-reactive WT clones have a modest version of the same phenotype, Fig 8g, h; p=0.006).

Altogether, I consider that the paper provides a wealth of quality data, unifying, amongst other things, findings from various other studies into a quite coherent frame work. Even though some of the findings, mostly the anergic phenotype T cells, are not completely explained, they are important to be shared with the field.